# HIV-1 nuclear import in macrophages is regulated by CPSF6-capsid interactions at the nuclear pore complex

David Alejandro Bejarano[1†], Ke Peng[1†‡*], Vibor Laketa[1,2], Kathleen Börner[1,2], K Laurence Jost[1,2], Bojana Lucic[2,3], Bärbel Glass[1], Marina Lusic[2,3], Barbara Müller[1], Hans-Georg Kräusslich[1*]

[1]Department of Infectious Diseases Virology, University of Heidelberg, Heidelberg, Germany; [2]German Center for Infection Research, Heidelberg, Germany; [3]Department of Infectious Diseases, Integrative Virology, University of Heidelberg, Heidelberg, Germany

*For correspondence:
pengke@wh.iov.cn (KP);
hans-georg.kraeusslich@med.uni-heidelberg.de (H-GKä)

†These authors contributed equally to this work

Present address: ‡State Key Laboratory of Virology, Wuhan Institute of Virology, Chinese Academy of Sciences, Wuhan, China

Competing interests: The authors declare that no competing interests exist.

**Abstract** Nuclear entry of HIV-1 replication complexes through intact nuclear pore complexes is critical for successful infection. The host protein cleavage-and-polyadenylation-specificity-factor-6 (CPSF6) has been implicated in different stages of early HIV-1 replication. Applying quantitative microscopy of HIV-1 reverse-transcription and pre-integration-complexes (RTC/PIC), we show that CPSF6 is strongly recruited to nuclear replication complexes but absent from cytoplasmic RTC/PIC in primary human macrophages. Depletion of CPSF6 or lack of CPSF6 binding led to accumulation of HIV-1 subviral complexes at the nuclear envelope of macrophages and reduced infectivity. Two-color stimulated-emission-depletion microscopy indicated that under these circumstances HIV-1 complexes are retained inside the nuclear pore and undergo CA-multimer dependent CPSF6 clustering adjacent to the nuclear basket. We propose that nuclear entry of HIV-1 subviral complexes in macrophages is mediated by consecutive binding of Nup153 and CPSF6 to the hexameric CA lattice.
DOI: https://doi.org/10.7554/eLife.41800.001

## Introduction

Reverse transcription of the viral single-stranded RNA genome into double-stranded cDNA, followed by integration into the host genome, are defining features of retrovirus replication (*Hu and Hughes, 2012*). These events occur in the cytoplasm and nucleus of the newly infected cell within ill-defined nucleoprotein complexes, termed reverse transcription (RTC) and pre-integration complex (PIC), respectively. Accumulating evidence indicates that the viral capsid structure plays an essential role in early replication (*Campbell and Hope, 2015*; *Yamashita and Engelman, 2017*). Mutations in the HIV-1 CA (capsid) protein have been shown to cause defects in post-entry stages, and various cellular proteins binding CA exhibit positive or negative effects on early HIV-1 replication (*Hilditch and Towers, 2014*; *Ambrose and Aiken, 2014*; *Campbell and Hope, 2015*; *Yamashita and Engelman, 2017*). Accordingly, CA and/or the structure of the viral capsid have been implicated in reverse transcription, cytoplasmic trafficking, evasion from the cell-autonomous immune response and nuclear entry (*Ambrose and Aiken, 2014*; *Hilditch and Towers, 2014*; *Yamashita and Engelman, 2017*; *Campbell and Hope, 2015*).

Neither the exact composition of the HIV-1 RTC/PIC nor the functional role of host cell dependency or restriction factors that interact with CA during early replication is currently well understood. Furthermore, the time of capsid uncoating during early HIV-1 replication is not well established, and may differ depending on the target cell type (*Arhel et al., 2007*; *Zhou et al., 2011*; *Hulme et al.,*

**eLife digest** Viruses are miniscule parasites that hijack the resources of a cell to make more of themselves. For many, this involves getting inside the nucleus, the fortress that protects the cell's genetic information. To do so, viruses need to first find a way through a double-layered membrane called the nuclear envelope, which only opens up when a cell divides.

Yet, the human immunodeficiency virus type 1 (HIV-1) can infect cells that no longer divide, and in which the nucleus' walls never come down. The virus cores then head for the nuclear pores, heavily guarded holes in the nuclear envelope that allow the cell's own molecules to go in and out of the nucleus. But HIV-1 is too big to fit through, as its genetic information is encased in a capsid, a coat made of a complex assembly of proteins. However, research shows that these capsid proteins can bind to host proteins at the pore or even inside the nucleus. For example, the capsid protein can recognize the pore protein Nup153, or the nuclear protein CPSF6. These interactions could help the virus make its way in, but how these events unfold is still unclear.

To explore this, Bejarano, Peng et al. attached fluorescent labels to HIV-1 and watched as it infected non-dividing cells. Rather than completely get rid of their capsid before they crossed the pores, the virus particles hung on to a large part of their lattice. This remaining coat then attached to CPSF6; when this protein was missing or could not bind to capsid proteins, the viral complexes got stuck in the nuclear pores. This suggests that the capsid lattice could first interact with Nup153 inside the pores: then, CPSF6 would take over, knocking Nup153 away and pulling HIV-1 into the nucleus.

Armed with this knowledge, virologists and drug developers could try to block HIV-1 from entering the cell's nucleus; they could also start to dissect how drugs that target the HIV-1 capsid work. Ultimately, HIV-1 may serve as a model to unravel how large objects can pass the nuclear pore, which may help us understand how molecules are constantly trafficked in and out of the nucleus.

DOI: https://doi.org/10.7554/eLife.41800.002

*2015*; *Chen et al., 2016*). Individual viruses entering a host cell may follow different - productive and non-productive – pathways, and it is crucial to identify productive phenotypes correlating with infection. It is generally accepted that incoming capsids remain intact after cytoplasmic entry and during the initial stages of reverse transcription, while different reports suggest uncoating in the cytoplasm or at the nuclear pore complex (NPC) (e.g. *Xu et al., 2013*; *Mamede et al., 2017*; *Francis and Melikyan, 2018*). Nuclear HIV-1 PIC have been observed to contain at least some CA (*Zhou et al., 2011*; *Peng et al., 2014*; *Hulme et al., 2015*; *Chin et al., 2015*; *Chen et al., 2016*; *Stultz et al., 2017*), but the amount of residual CA and whether it maintains the lattice structure are unknown. Furthermore, the size of the conical viral capsid (~60 nm at the broad end (*Briggs et al., 2003*)) is larger than the width of the NPC transport channel (*Beck et al., 2004*), indicating that (partial) capsid disintegration or at least capsid or NPC remodeling is needed for HIV-1 nuclear translocation in non-dividing cells.

Cleavage and polyadenylation specificity factor 6 (CPSF6) was initially identified as an HIV-1 restriction factor when exogenously expressed in a truncated form. The truncated protein is enriched in the cytoplasm, interacts with HIV-1 CA and impairs HIV-1 replication prior to nuclear import (*Lee et al., 2010*). In contrast, neither overexpression of full length CPSF6, which is almost exclusively nuclear, nor knock-down of CPSF6 significantly affected HIV-1 infectivity in cell lines (*Lee et al., 2010*; *Hori et al., 2013*; *Fricke et al., 2013*; *De Iaco et al., 2013*). Structural analyses identified a CPSF6-interacting interface and an overlapping site interacting with the nucleoporin Nup153 on the hexameric form of CA, the basic building block of the viral capsid (*Price et al., 2012*; *Price et al., 2014*; *Bhattacharya et al., 2014*). An HIV-1 derivative carrying a point mutation within this CPSF6-binding motif in CA (N74D) exhibited impaired CPSF6 binding in vitro and escaped restriction by truncated CPSF6, but displayed full infectivity in reporter cell lines (*Lee et al., 2010*; *Schaller et al., 2011*; *Ambrose et al., 2012*). In contrast, both CPSF6 knock-down and the N74D exchange impaired HIV-1 infection in post-mitotic primary human macrophages (*Schaller et al., 2011*; *Ambrose et al., 2012*), and this effect was attributed to induction of an

interferon response triggered by viral DNA sensing (*Rasaiyaah et al., 2013*). Based on these results, it was speculated that cytoplasmic CPSF6 binds incoming viral capsids and blocks reverse transcription during cytoplasmic transport to prevent recognition of newly synthesized cDNA by cytoplasmic DNA sensors (*Rasaiyaah et al., 2013*).

More recently, CPSF6 was suggested to affect HIV-1 nuclear entry in a HeLa-based reporter cell line (*Chin et al., 2015*) and to be an important factor in targeting HIV-1 integration in infected primary CD4$^+$ T-cells and macrophages (*Sowd et al., 2016*; *Rasheedi et al., 2016*; *Achuthan et al., 2018*). Strikingly, a later study described another point mutation in the CPSF6-binding interface of HIV-1 CA (A77V), which also abolished interaction with CPSF6 but did not appear to affect replication in primary cells (i.e. primary human macrophages and CD4$^+$ T-cells), although it was negatively selected after passaging in vivo (*Saito et al., 2016b*). Taken together, it is clear that CPSF6 has an important, CA dependent function in early HIV-1 infection of macrophages, which may not be fully recapitulated in HeLa- or 293T-based reporter cell lines. This may reflect differences between cell lines and primary cells, but also between different cell types. The mechanism(s) of action of CPSF6 is not fully defined, and the relative contribution of cytoplasmic versus nuclear CPSF6 as well as the reason for the apparent cell-type specific differences are not resolved.

We have previously established a microscopy-based approach for quantitative analysis of HIV-1 post entry events on a single particle level (*Peng et al., 2014*). Labeling nascent RT products by incorporation of the thymidine analog EdU followed by click labeling, in conjunction with fluorescent labeling of the *bona fide* RTC/PIC component IN, identified reverse transcription competent HIV-1 RTC/PIC in the cytoplasm and nucleus of infected cells and enabled direct visualization of viral and cellular proteins associated with these complexes. Employing this system to investigate CPSF6 recruitment, we had observed weak or no CPSF6 signals on cytosolic RTC/PIC in model cell lines; pronounced-co-localization was only observed when transportin 3 (TNPO3), which is needed for CPSF6 nuclear import, was depleted (*Peng et al., 2014*). We have now used this approach for a detailed analysis of CPSF6 recruitment and its role for HIV-1 nuclear import in primary human monocyte-derived macrophages (MDM). CPSF6 was strongly enriched on nuclear complexes, and depletion of CPSF6 or the A77V mutation in CA reduced HIV-1 infectivity in MDM. RTC/PIC accumulated close to the nuclear envelope in these cases. Two-color Stimulated Emission Depletion (STED) microscopy revealed that CA-containing HIV-1 complexes directly co-localized with NPCs, and CPSF6 was associated with the nuclear basket at these sites in a CA-dependent manner. These results indicate that CPSF6 facilitates nuclear entry of HIV-1 in post-mitotic human macrophages in a CA–dependent manner at the level of the NPC.

## Results

### CPSF6 binding of the RTC/PIC does not impair reverse transcription

The poor association of cytoplasmic RTC/PIC with CPSF6 observed in our previous study (*Peng et al., 2014*) argued against the model that CPSF6 regulates viral reverse transcription during cytoplasmic trafficking (*Rasaiyaah et al., 2013*). Our experimental system allowed us to directly address this issue by correlating the presence of CPSF6 on cytosolic RTC/PIC with the EdU/click signal intensity as a measure of reverse transcription products. These experiments were performed in a HeLa-derived TNPO3 knock-down cell line which displays a high cytosolic level of CPSF6 (*Thys et al., 2011*). Cells were infected with HIV-1 carrying IN.eGFP as RTC/PIC marker, subjected to EdU incorporation, and fixed and click-labeled 4.5 hr post infection. IN.eGFP/EdU positive objects were classified according to whether or not they were associated with CPSF6 immunofluorescence. In accordance with our previous results (*Peng et al., 2014*), the majority of cytoplasmic RTC/PIC (95/121; 78.5%) was positive for CPSF6 in this cell line with high cytoplasmic CPSF6 levels (*Figure 1—figure supplement 1A*). EdU signal intensities on individual CPSF6-positive complexes were found to be significantly higher on average compared to those on CPSF6-negative, but IN.eGFP-positive objects (*Figure 1—figure supplement 1B*), implying that CPSF6 association with cytoplasmic RTC/PIC did not inhibit reverse transcription. To analyze further whether CPSF6 affects reverse transcription, CPSF6 was depleted in MDM followed by infection and RTC/PIC quantification. The number of reverse-transcription competent RTC/PIC (i.e. EdU positive signals co-localizing with IN.eGFP) in CPSF6 depleted MDM was comparable with that of control cells (*Figure 1—figure supplement 1C*).

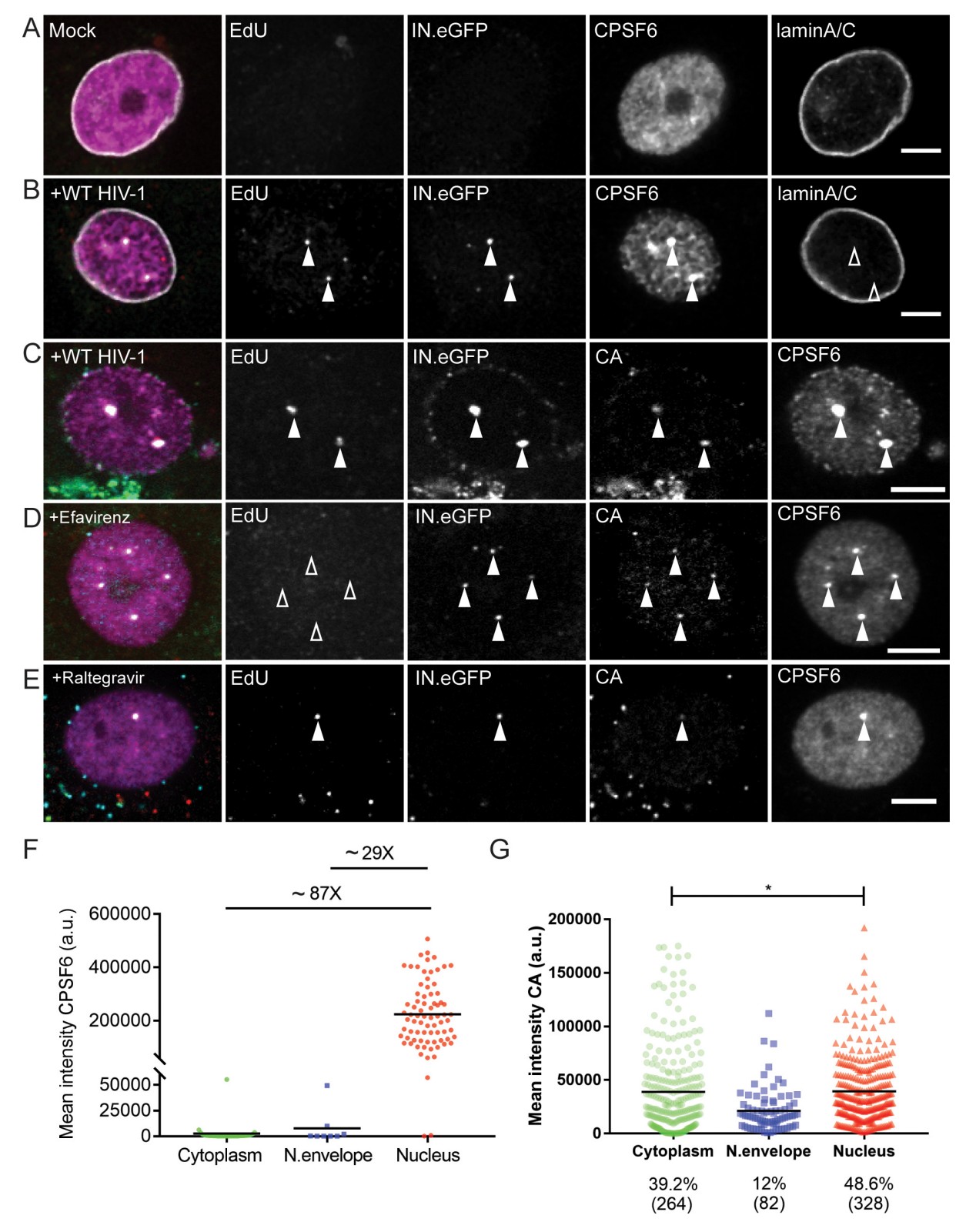

**Figure 1.** CPSF6 is enriched on nuclear HIV-1 complexes in infected macrophages. MDM were mock-infected (**A**) or infected with HIV-1 $_{NL4-3}$4059 (IN. eGFP) at an MOI of 14.5 (**B–G**) in the presence of 10 µM EdU (**A–G**), and 5 µM Efavirenz (**D**) or 5 µM Ral (Raltegravir) (**E**) for 48 hr, click-labeled and immunostained with anti-CPSF6 antibody (magenta). Z-stacks were acquired by spinning disk confocal microscopy (SDCM). RTC/PIC were identified *via* co-localization of IN.eGFP (green) and EdU (red) signals. Images show representative z-sections of the nucleus of infected cells. Solid arrowheads

*Figure 1 continued on next page*

*Figure 1 continued*

indicate nuclear complexes. Open arrowheads indicate lack of co-localization for the respective stain. Scale bars represent 5 µm. (**A, B**) Distribution of CPSF6 in the nucleus of a mock-infected or infected cell. Immunostaining against lamin A/C was performed to visualize the nuclear envelope. (**C–E**) As in A,B, but immunostaining against CA (cyan) was performed instead of lamin staining. (**F**) Distribution and mean intensities of CPSF6 signals co-localizing with individual RTC/PIC in the cytoplasm, close to the nuclear envelope and inside the nucleus; each dot represents one RTC/PIC. The graph summarizes data from two donors from two independent experiments. 62 cells were analyzed in total. (**G**) Distribution and mean intensities of CA signals co-localizing with individual EdU-positive RTC/PIC. Numbers of total RTC/PIC detected are given in parentheses. The graph summarizes data from four donors from three independent experiments. 92 cells were analyzed in total. Statistical significance was assessed with two-tailed non-paired Mann-Whitney test; *p=0.013.

DOI: https://doi.org/10.7554/eLife.41800.003

The following source data and figure supplements are available for figure 1:

**Source data 1.** Mean CPSF6 and CA signal intensities of individual HIV-1 EdU positive subviral complexes at different subcellular localizations.
DOI: https://doi.org/10.7554/eLife.41800.007
**Figure supplement 1.** Effects of cytoplasmic CPSF6 on HIV-1 reverse transcription.
DOI: https://doi.org/10.7554/eLife.41800.004
**Figure supplement 2.** Replication kinetics of HIV-1 in primary macrophages and induction of CPSF6 enrichment by lentiviral transduction.
DOI: https://doi.org/10.7554/eLife.41800.005
**Figure supplement 3.** Effect of CPSF6 knockdown on CA signals.
DOI: https://doi.org/10.7554/eLife.41800.006
**Figure supplement 3—source data 1.** Data includes mean CA signals of individual WT and A77V HIV-1 subviral complexes at different subcellular localization after CPSF6 knock-down.
DOI: https://doi.org/10.7554/eLife.41800.008

These results are inconsistent with an inhibitory effect of cytoplasmic CPSF6 on HIV-1 reverse transcription. We therefore focused on the characterization of CPSF6 in the nucleus and at the NPC in primary human MDM in the following experiments.

## CPSF6 is strongly recruited to nuclear subviral HIV-1 complexes in infected macrophages

Reverse transcription of HIV-1 in MDM is much slower than in reporter cell lines or activated T-cells, presumably due to the low dNTP levels in these non-dividing cells (*Diamond et al., 2004*). We therefore performed inhibitor time-of-addition experiments to define the appropriate time window for RTC/PIC detection in this cell type. MDM were prepared from healthy blood donors and infected with HIV-1 carrying an R5-tropic Env protein derived from a primary human isolate (*Schnell et al., 2011*). The non-nucleosidic reverse transcriptase (RT) inhibitor efavirenz (EFV) or DMSO was added at different time points post single-round infection (p.i.), and the percentage of infected cells was determined by CA immunostaining 6d p.i.. Only minor resilience against EFV inhibition was observed at 24 h p.i. and this increased until 72 h p.i., where EFV inhibition was lost (*Figure 1—figure supplement 2A,B*). These results confirmed that completion of reverse transcription occurs late in MDM (supporting a recent more detailed analysis of replication dynamics (*Bejarano et al., 2018*)) and defined the time window for detection of HIV-1 RTC/PIC.

MDM were infected with R5-tropic HIV-1 carrying IN.eGFP followed by detection of RTC/PIC *via* click-labeling of EdU incorporated into nascent viral DNA. The nuclear envelope was visualized by immunostaining of lamin A/C. At 48 h p.i., mock-infected cells displayed diffuse, predominantly nuclear localization of CPSF6 (*Figure 1A*), as previously reported (*Dettwiler et al., 2004*). In contrast, CPSF6 was strongly enriched on almost all nuclear HIV-1 complexes (75/77; 97.4%) in infected MDM, and strong punctate CPSF6 signals co-localizing with nuclear HIV-1 complexes were easily detected above the diffuse nuclear background (*Figure 1B*). Since the signal intensities of HIV-1 associated CPSF6 punctae were clearly much higher than those of the small nuclear CPSF6 speckles observed in both non-infected and infected cells, we consider it likely that this reflects CPSF6 recruitment by subviral complexes rather than recruitment of HIV-1 derived structures to pre-existing CPSF6 clusters. CPSF6 signal intensities were much higher on nuclear HIV-1 subviral complexes compared to those on complexes localized near the nuclear envelope. CPSF6 levels on the latter structures were generally close to the detection level, and cytoplasmic RTC/PIC were almost always CPSF6-negative (*Figure 1F*). Interestingly, cells transduced with lentiviral vectors expressing a non-

targeted shRNA exhibited similar characteristic CPSF6 clusters in the nucleus (*Figure 1—figure supplement 2C*), while this was not observed in mock-transduced cells. We attribute these signals to nuclear complexes of the lentiviral vector. These results clearly show that CPSF6 association with RTC/PIC occurs mainly in the nucleus in MDM and leads to strong CPSF6 clustering on the subviral complex.

To assess the degree of CA retention on nuclear subviral complexes, MDM were infected with HIV-1 for 48 hr as above and subjected to CA immunostaining. In agreement with our previous findings (*Peng et al., 2014*), nuclear HIV-1 RTC/PIC were strongly CA-positive, with CA signals clearly co-localizing with EdU, IN.eGFP and CPSF6 (*Figure 1C*). Quantifying localization and CPSF6/CA association of HIV-1 complexes in a total of 92 infected MDM at 48 h p.i., we observed a large majority of IN.eGFP positive structures outside the nucleus (7,904/8,403; 94%). These structures mostly lacked detectable EdU signals (7558/7904; 95.6%), in agreement with the assumption that the majority of particles in the cytosolic area represent virions taken up into endosomes (inaccessible to EdUTP) and non-productive particles. In contrast, the majority of IN.eGFP positive objects within the nucleus were EdU positive (328/499; 65.7%). The observation that a subset of complexes detected in the nucleus was EdU negative confirms that reverse transcription is not a prerequisite for nuclear import of HIV-1 complexes, as reported by (*Burdick et al., 2017*). In a recent study on HIV-1 replication dynamics in macrophages, we observed higher EdU intensities associated with nuclear complexes compared to complexes near the nuclear envelope, however, suggesting that reverse transcription may promote efficiency of nuclear entry or that reverse transcription can even be completed in the nucleus in this cell type (*Bejarano et al., 2018*). 325 of the 328 EdU-positive complexes in the nucleus (99%) were associated with detectable CA signals, and a similar CA signal was also observed on 85% of the EdU-negative nuclear IN.eGFP-positive complexes (146/171).

Mean CA signal intensity of individual RTC/PIC differed only modestly between cytoplasmic and nuclear CA-positive HIV-1 structures, suggesting that nuclear RTC/PIC in HIV-1 infected MDM retain the majority of CA molecules. Interestingly, the mean CA signal intensity associated with RTC/PIC close to the nuclear envelope was lower than observed for either cytoplasmic or nuclear RTC/PIC (*Figure 1G*). To test whether recruitment of CPSF6 to HIV-1 nuclear complexes affects CA retention, MDM were subjected to CPSF6 knock-down prior to HIV-1 infection. CPSF6 knock-down had no apparent effect on CA signals associated with HIV-1 RTC/PIC independent of the subcellular localization (*Figure 1—figure supplement 3A*). Comparable results were obtained upon infection of MDM with CPSF6 binding-defective HIV-1 carrying the A77V exchange in CA (*Figure 1—figure supplement 3B*).

To test whether CPSF6 enrichment on nuclear subviral complexes requires HIV-1 reverse transcription or integration, MDM were infected and labeled in the presence of RT or IN inhibitors, respectively. Nuclear import of HIV-1 subviral complexes and CPSF6 recruitment to these structures was independent of reverse transcription (*Figure 1D*; 102/105 nuclear complexes were CPSF6-positive; 97.1%) and integration (*Figure 1E*: 101/104 nuclear complexes were CPSF6-positive; 97.1%).

## Characterization of CPSF6 and LEDGF recruitment to nuclear HIV-1 complexes

To verify that the CPSF6-enriched nuclear structures contain reverse-transcribed HIV-1 genomes, we performed immunofluorescence combined with DNA in situ hybridization (immuno-FISH) on infected MDM at 72 h p.i.. Co-localization of viral DNA signals with CA immunostaining and CPSF6-enrichment (*Figure 2A*) confirmed that the EdU signal on nuclear complexes corresponds to HIV-1 DNA and ascertained that the identified objects represent HIV-1 replication complexes that have undergone reverse transcription.

We then investigated whether a specific CPSF6 complex is recruited to nuclear HIV-1 RTC/PIC. CPSF6 is part of the cellular CF Im complex involved in pre-mRNA processing (*Gruber et al., 2012*; *Hardy and Norbury, 2016*). Two forms of this tetrameric complex have been described. They are composed of two CPSF5 molecules, associated with two molecules of either CPSF6 or CPSF7 (*Gruber et al., 2012*). MDM were infected with HIV-1 for 48 hr and co-immunostained with antibodies against CPSF6 and CPSF5, or against CPSF7 and CPSF5. We observed clear co-localization of CPSF5 and CPSF6 on almost all nuclear complexes analyzed (42/44), while CPSF7 was only detected on one of 82 nuclear complexes analyzed (*Figure 2B*). Thus, we conclude that the CPSF5[2]-CPSF6[2] form of the CF Im complex is recruited to nuclear HIV-1 complexes.

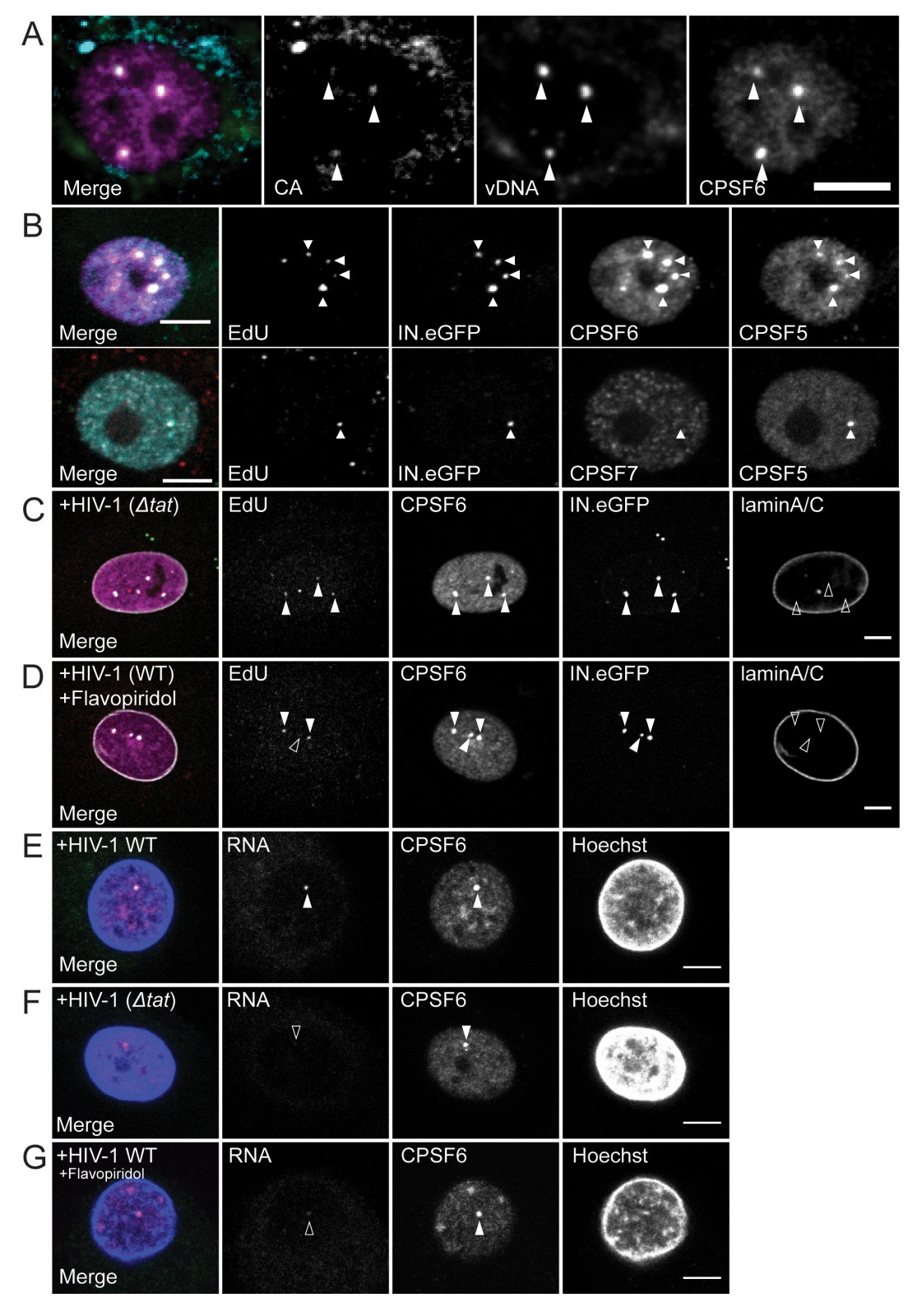

**Figure 2.** Characterization of the composition of nuclear HIV-1 replication complexes. (**A**) MDM from three donors were infected with HIV-1 $_{NL4-3}$4059 at an MOI of 8 for 72 hr. Viral DNA (green) was detected by FISH as described in Materials and Methods. CPSF6 (magenta) and CA (cyan) were detected by immunostaining and images were recorded by SDCM. The maximum projection of three focal planes acquired with an axial spacing of 0.2 μm is shown. Arrowheads indicate nuclear complexes. (**B**) MDM from three donors were infected with 100ng p24 (MOI 14.5) HIV-1 $_{NL4-3}$4059 (IN.eGFP) for 48

*Figure 2 continued on next page*

*Figure 2 continued*

hr, fixed and click-labeled as in *Figure 1*. CPSF5 (cyan) and CPSF6 (magenta) (top panel) or CPSF7 (white) and CPSF5 (cyan) (bottom panel) were detected by immunostaining. Images show a section through the nuclei of infected cells. Arrowheads indicate nuclear complexes. 112 cells were analyzed in total. (C,D) MDM from two donors in two independent experiments were infected with 100ng p24 (MOI 14.5) HIV-1 $_{NL4-3\Delta Tat}$4059 (IN.eGFP) (C) or HIV-1 $_{NL4-3}$4059 (IN.eGFP) (D). At 96 h p.i., a final concentration of 5 μM Flavopiridol (D) was added to the medium and infection was continued for 12 hr. Cells were fixed, click-labeled and CPSF6 (magenta) and laminA/C (white) were detected by immunostaining. Images show a section through the nucleus of representative infected cells. Arrowheads indicate nuclear complexes. 22 cells were analyzed in (C) and 37 cells in (D). (E,F) MDM from three donors, in two independent experiments, were infected at an MOI of 8 with HIV-1 $_{NL4-3}$4059 (E) or HIV-1 $_{NL4-3\Delta Tat}$4059 (F). At 108 h p.i. cells were fixed. CPSF6 (magenta) was detected by immunostaining and nucleus (blue) with Hoechst. Viral RNA (green) was detected by RNA FISH as explained in Materials and Methods. Arrowhead indicates nuclear complexes. 42 cells were analyzed in total in (E) and 49 cells in (F). (G) MDM from the same donors as in (E,F) were infected with HIV-1 $_{NL4-3}$4059 using the same conditions. At 96 h p.i. 5 μM Flavopiridol was added to the medium for 12 hr. Viral RNA (green) was detected by RNA FISH. CPSF6 (magenta) was detected by immunostaining and nucleus (blue) with Hoechst. Arrowheads indicate nuclear CPSF6 enrichments. 36 cells were analyzed in total. Scale bars in A-G: 5 μm.

DOI: https://doi.org/10.7554/eLife.41800.009

The following figure supplements are available for figure 2:

**Figure supplement 1.** Inhibition of HIV-1 transcription and LEDGF localization.

DOI: https://doi.org/10.7554/eLife.41800.010

**Figure supplement 2.** FISH analysis of cells infected with a CPSF6-binding defective HIV-1 variant.

DOI: https://doi.org/10.7554/eLife.41800.011

Given the role of the CF Im complex in cellular pre-mRNA processing, we investigated whether CPSF6 accumulation on nuclear HIV-1 complexes depends on their transcriptional activity. To this end, MDM were infected with a transcriptionally impaired HIV-1 variant carrying a deletion in the viral Tat transactivator (HIV-1$_{NL4-3\Delta Tat}$). Alternatively, wild-type HIV-1 infected MDM were treated at 96 h p.i. with the transcription inhibitor flavopiridol for 12 hr. HIV-1 transcription and infectivity were strongly impaired by either, Tat deletion or flavopiridol treatment (*Figure 2—figure supplement 1A and B*). Impairment of HIV-1 RNA transcription did not affect CPSF6 recruitment to nuclear HIV-1 complexes, however. CPSF6 enrichment on nuclear replication complexes of HIV-1$_{NL4-3\Delta Tat}$ infected MDM (54/58; 93%; *Figure 2C*) was comparable with wild-type HIV-1, and this was also true for flavopiridol treatment (63/64; 98%; *Figure 2D*). CPSF6 recruitment to transcriptionally inactive nuclear HIV-1 complexes was confirmed by immunostaining in combination with RNA FISH at 108 h p.i.. This analysis revealed that CPSF6-positive nuclear punctae in wild-type HIV-1 infected MDM mostly co-localized with viral RNA signals (50/69; 72.5%; *Figure 2E*), indicating some transcriptional activity. On the other hand, nuclear CPSF6 punctae in MDM infected with the Tat-defective HIV-1 variant (72/98; 74%; *Figure 2F*) or in the presence of Flavopiridol (29/34; 85%; *Figure 2G*) mostly lacked detectable FISH signals for viral RNA. These results indicate that recruitment of CPSF6 to HIV-1 nuclear replication complexes does not require active transcription of the HIV-1 genome and does thus not result from CPSF6 function in pre-mRNA processing.

The p75 isoform of the host cell factor Lens Epithelium-Derived Growth Factor (LEDGF/p75) has been reported to be important for HIV-1 integration and to influence integration site selection (*Kvaratskhelia et al., 2014*; *Debyser et al., 2015*). We therefore analyzed nuclear HIV-1 complexes for the presence of LEDGF. Co-immunostaining of HIV-1 infected MDM with antibodies against LEDGF (detecting both the p75 and p52 isoform) and CPSF6 revealed all nuclear complexes to be CPSF6-positive in this experiment and ~80% to contain detectable LEDGF as well (*Figure 2—figure supplement 1C*).

## CPSF6 is important for HIV-1 infectivity in macrophages

In order to determine whether enrichment of CPSF6 on nuclear HIV-1 complexes is functionally relevant for HIV-1 replication, we performed CPSF6 knock-down experiments and employed a CPSF6 binding deficient virus. Mutation A77V in CA has previously been reported to impair CPSF6 interaction without affecting replication in MDM (*Saito et al., 2016b*), and thus to be more specific compared to the N74D mutation used in other studies. MDM from three donors each were transduced with either lentiviral or adeno-associated virus (AAV) based vectors expressing a combination of three shRNAs against CPSF6 or a non-targeted shRNA. CPSF6 signal intensities in the nucleus were quantitated to determine the level of knock-down (*Figure 3—figure supplement 1A*). While both

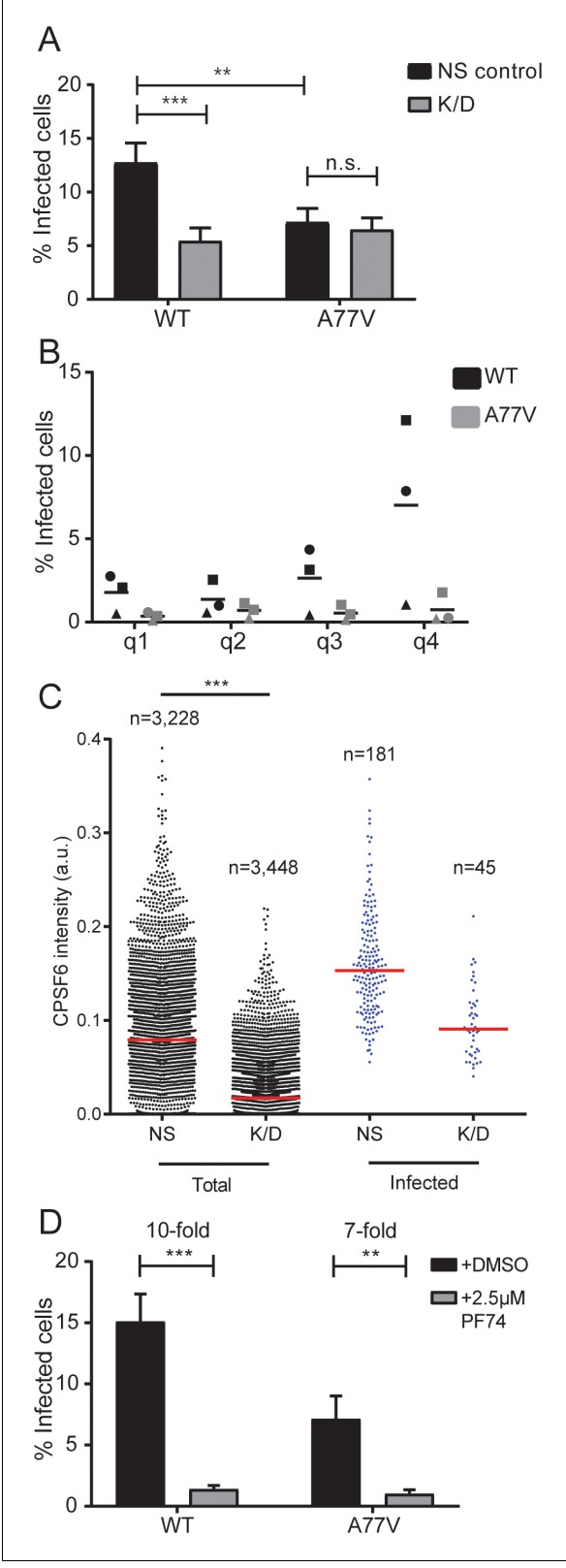

**Figure 3.** Effect of CPSF6 depletion on HIV-1 infection in primary macrophages. (**A**) MDM from six donors in three independent experiments were transduced with either lentiviral vectors or AAV expressing a non-targeted shRNA (NS control) or shRNAs against CPSF6 (K/D) as described in Materials and Methods. Twelve days after initial transduction, cells were infected with 15 ng CA (MOI 3.5) of HIV-1 $_{NL4-3}$ WT or A77V. To prevent secondary
*Figure 3 continued on next page*

*Figure 3 continued*

infection, 5 µM Maraviroc was added to the medium at 24 h p.i. The proportion of infected cells was determined by immunostaining using antiserum against CA at 6d p.i. Three independent experiments each, using cells from different donors, were conducted for lentivirally and AAV transduced cells, and infections were performed in triplicate per condition in each experiment. The graph shows mean values and SEM for the proportion of infected cells from all data sets. Statistical significance was assessed with two-tailed non-paired Mann-Whitney test; **p=0.0008, ***p<0.0001, n.s.: not significant, a.u.: arbitrary unit. Please refer to *Figure 3—figure supplement 1* for data sets from individual experiments. (**B**) Correlation between CPSF6 levels and infectivity. MDM were transduced with AAV and infected as in (**A**). CPSF6 signal intensities from individual infected and non-infected cells were quantified as described in Materials and Methods. CPSF6 intensities per cell from six randomly selected wells were used to separate the population into quartiles (q1 <62. q2 <95. q3 <117. q4 <353 a.u.) and the proportion of infected cells was determined for each quartile. Symbols represent mean values calculated for samples from three individual donors; lines indicate the mean value from each data set. For each condition,>10,000 cells were analyzed, per donor. (**C**) MDM were transduced with lentiviral vectors and infected with HIV-1 $_{NL4-3}$ 4059-WT, as described in (**A**). CPSF6 signal intensities from individual infected and non-infected cells were quantified as described in Materials and Methods. The graph shows CPSF6 intensity values from n individual NS or K/D cells for the total (left) and infected (right) cell population. Median values are indicated by red lines. Statistical significance was assessed with two-tailed non-paired Mann-Whitney test; ***: p<0.0001. a.u.: arbitrary unit. (**D**) MDM were infected with HIV-1 $_{NL4-3}$ WT or A77V, as described above, in the presence of 2.5 µM PF74. At 24 h p.i. 5 µM Maraviroc was added to the medium. Cells were fixed 6d p.i. and the proportion of infected cells was determined by immunostaining using antiserum against CA. Two independent experiments, using cells from four donors, were performed. Infections were performed in triplicate per condition in each experiment. The graph shows mean values and SEM for the proportion of infected cells from all data sets. Statistical significance was assessed with two-tailed non-paired Mann-Whitney test; **p=0.0013, ***p<0.0001, n.s.: not significant.

DOI: https://doi.org/10.7554/eLife.41800.012

The following source data and figure supplements are available for figure 3:

**Source data 1.** Effect of CPSF6 depletion on HIV-1 infectivity in primary macrophages.
DOI: https://doi.org/10.7554/eLife.41800.015
**Figure supplement 1.** Knockdown efficiency and HIV-1 infectivity in primary macrophages from different donors.
DOI: https://doi.org/10.7554/eLife.41800.013
**Figure supplement 1—source data 1.** Mean CPSF6 signal intensities of individual cells from multiple donors after CPSF6 knock-down.
DOI: https://doi.org/10.7554/eLife.41800.016
**Figure supplement 2.** Relative infectivity of the N74D CA variant.
DOI: https://doi.org/10.7554/eLife.41800.014
**Figure supplement 2—source data 2.** Raw infectivity data of primary macrophages from multiple donors infected with N74D HIV-1.
DOI: https://doi.org/10.7554/eLife.41800.017

approaches yielded ca. 50–60% reduction of mean CPSF6 signal intensities (*Figure 3—figure supplement 1A*, compare top panels to bottom panels), transduction with lentiviral vectors resulted in the appearance of nuclear CPSF6 punctae (*Figure 1—figure supplement 2C*) indistinguishable from those associated with HIV-1 derived complexes. We therefore employed AAV vectors for all imaging experiments, while lentiviral knock-down was used in some infection experiments as well.

Transduced MDM were infected with non-labeled R5-tropic HIV-1 in a single-round infection at an MOI of 3.5 (based on titration in HeLa-based reporter cells) and scored for productive infection at day 6 p.i.. Knock-down of CPSF6 modestly, but significantly reduced wild-type HIV-1 infection (*Figure 3A*). A similar decrease in infectivity was seen for the A77V variant without CPSF6 silencing, different from a previous report where this variant appeared to be unimpaired (*Saito et al., 2016b*). CPSF6 depletion had no additional effect on infection by the A77V variant, indicating that its phenotype is indeed due to loss of CPSF6 binding (*Figure 3A* and *Figure 3—figure supplement 1B*, bottom panels show data for AAV mediated depletion). Since CPSF6 knock-down levels varied between individual cells, we analyzed the correlation between CPSF6 signal intensity and HIV-1 infection at the single-cell level (*Figure 3B,C*). Stratification of cells into quartiles according to CPSF6 staining intensity revealed a correlation between HIV-1 infection and CPSF6 staining intensity for wild-type HIV-1, but not for the A77V variant (*Figure 3B*). Single-cell analysis further revealed that a certain threshold level of CPSF6 was apparently required for MDM to become productively infected

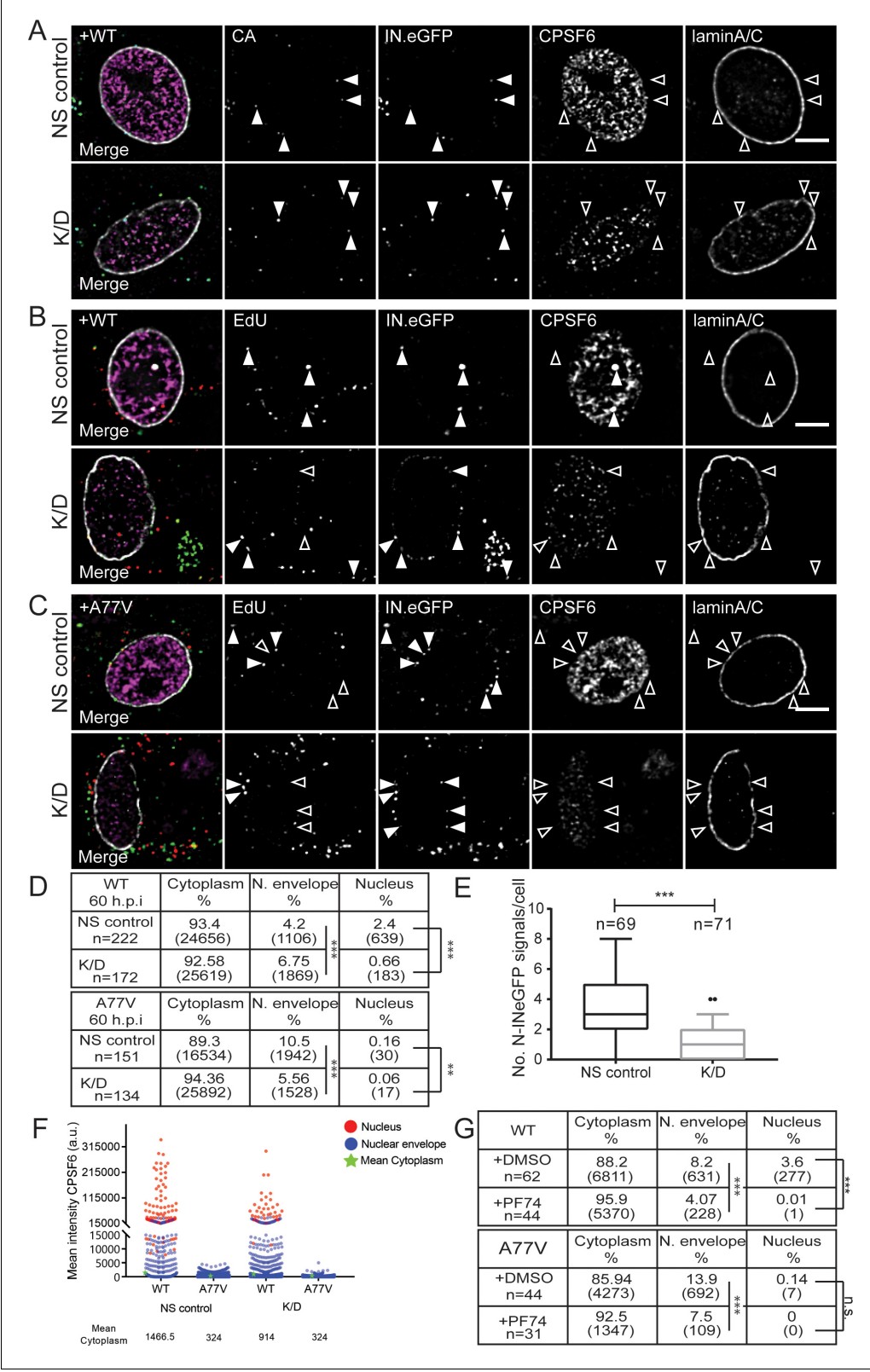

**Figure 4.** Effect of CPSF6 knock-down on nuclear entry. MDM were transduced with AAVs as described in Materials and Methods. Subsequently, cells were infected with 100 ng CA (MOI 14.5) of HIV-1 $_{NL4-3}$ WT (IN.eGFP) (**A, B**) or A77V (IN.eGFP) (**C**) in the presence of 10 µM EdU. (**A**) At 24 h p.i., cells were fixed, and immunostained for CA (cyan), CPSF6 (magenta) and laminA/C (white). Filled arrowheads indicate IN.eGFP signals co-localizing with CA in cells displaying WT (NS control) or low (K/D) CPSF6 signal intensity. Open arrowheads indicate no co-localization for the respective marker. See

*Figure 4 continued on next page*

*Figure 4 continued*

*Figure 4—figure supplement 1* for corresponding data obtained with HIV-1$_{NL4-3}$ A77V. **(B, C)** At 60 h p.i., cells were fixed, click-labeled and immunostained against CPSF6 (magenta) and laminA/C (white). Solid arrowheads indicate IN.eGFP positive objects co-localizing with EdU in infected cells displaying WT (NS control) or low (K/D) CPSF6 signal intensity. Open arrowheads indicate lack of co-localization for the respective marker. Images in A-C show representative z-sections through the nuclear region of infected cells. IN.eGFP and EdU signals in the merged panels are represented in green and red, respectively. **(D)** Analysis of individual objects detected in cells from four donors at 60 h p.i. Tables summarize the subcellular distribution of IN.eGFP positive objects in n cells from four donors in three independent experiments, infected with HIV-1 $_{NL4-3}$ WT (IN.eGFP) or the A77V variant (IN.eGFP) at 60 h p.i. with or without knock-down of CPSF6. Numbers of detected objects are given in parentheses. Proportions of nuclear or close to the nuclear envelope IN.eGFP objects with and without CPSF6 knock-down were compared using a two-tailed Z-test ($\alpha = 0.05$); **: p=0.0002. ***: p<0.0001. **(E)** Numbers of nuclear IN.eGFP positive objects per cell in WT HIV-1 infected cells displaying normal (NS control) or low (K/D) CPSF6 signal intensity. Whiskers represent 5 and 95 percentile. Statistical significance was assessed with two-tailed non-paired Mann Whitney test; ***: p<0.0001. **(F)** Mean intensities of CPSF6 signals co-localizing with individual IN.eGFP positive objects at the indicated subcellular localization in cells infected with HIV-1 $_{NL4-3}$ WT (IN.eGFP) or the A77V variant (IN.eGFP) and displaying normal (NS control) or low (K/D) CPSF6 signal intensity (related to panel D). **(G)** Analysis of individual objects detected in cells from two donors at 60 h p.i. Tables summarize the subcellular distribution of IN.eGFP positive objects in n cells from two donors infected with HIV-1 NL4-3 WT (IN.eGFP) or A77V (IN.eGFP) at 60 h p.i. in the presence of DMSO or 2.5 μM PF74. Numbers of detected objects are given in parentheses. Proportions of IN.eGFP objects in the absence or presence of PF74 located in the nucleus or close to the nuclear envelope were compared using a two-tailed Z-test ($\alpha = 0.05$); ***p<0.0001, n.s.: not significant. See *Figure 4—figure supplement 1* for corresponding imaging data.

DOI: https://doi.org/10.7554/eLife.41800.018

The following source data and figure supplements are available for figure 4:

**Source data 1.** Effect of CPSF6 knock-down on nuclear entry.
DOI: https://doi.org/10.7554/eLife.41800.020
**Figure supplement 1.** Effect of CPSF6 knockdown on trafficking of RTC/PIC to the nuclear envelope and effect of addition of PF74 on nuclear entry.
DOI: https://doi.org/10.7554/eLife.41800.019
**Figure supplement 1—source data 1.** Mean CPSF6 signal intensities of individual WT and A77V HIV-1 subviral complexes after 24 h p.i. at different subcellular localizations in cells under non-silencing or CPSF6 knock-down conditions (*Figure 4—figure supplement 1*).
DOI: https://doi.org/10.7554/eLife.41800.021

(*Figure 3C*). This phenotype was more obvious in CPSF6 depleted MDM, but exhibited significant donor-to-donor variation. Taken together, these observations suggest that a threshold level of CPSF6 is required for efficient infection of MDM with wild-type HIV-1, and that CPSF6 facilitates HIV-1 infection in a CA-dependent manner.

## CPSF6 is required for nuclear import of the RTC/PIC, but not for cytoplasmic trafficking to the nuclear pore

Having established that CPSF6 promotes HIV-1 infection of MDM, we next aimed to define the replication step affected. For this, we visualized viral particles in infected MDM at two time points after infection: 24 h p.i., when reverse transcription was still ongoing, and 60 h p.i., when reverse transcription was completed in most cells (*Figure 1—figure supplement 2A*). MDM were transduced with AAV vectors for CPSF6 knock-down and subsequently infected with wild-type HIV-1 or the A77V variant, both carrying IN.eGFP at an MOI of 14.5 (determined on HeLa-derived reporter cells).

At 24 h p.i., IN.eGFP-positive subviral complexes derived from wild-type HIV-1 were frequently detected near the nuclear envelope in cells displaying either normal (NS control) or low (K/D) CPSF6 levels (3 or 6.2% of IN.eGFP-positive objects, respectively; *Figure 4A*, *Figure 4—figure supplement 1B*). This was similar for cells infected with the A77V variant (5.8 or 4.4%, respectively; *Figure 4—figure supplement 1A,B*), indicating that CPSF6 plays no role in RTC/PIC trafficking to the nuclear envelope in MDM. A small fraction of IN.eGFP signals was already observed inside the nucleus for wild-type HIV-1 at this time (<0.5% of IN.eGFP positive objects), while only a single nuclear IN.eGFP object was detected in HIV-1 A77V infected cells (*Figure 4—figure supplement 1B*).

As expected, a higher proportion of IN.eGFP-positive complexes had entered the nucleus at 60 h p.i.. In MDM transduced with non-targeted vector, 2.4% of IN.eGFP positive objects were detected in the nucleus at this time for wild-type HIV-1 infection (*Figure 4B,D*). Cells depleted of CPSF6 displayed a significantly lower number of nuclear IN.eGFP signals (0.7%; *Figure 4B,D,E*), while the proportion of IN.eGFP-positive complexes close to the nuclear envelope was higher than in cells expressing normal levels of CPSF6 (*Figure 4D*). The proportion of nuclear IN.eGFP-positive objects remained low for cells infected with the A77V variant at 60 h p.i. (0.2%; *Figure 4C,D*), and was

further reduced upon CPSF6 knock-down (*Figure 4C,D*). Interestingly, a higher proportion of HIV-1 complexes was observed close to the nuclear envelope for the A77V variant at this later time point compared to wild-type HIV-1 (*Figure 4C,D*).

The reduction of nuclear complexes for the A77V variant was independently confirmed by immuno-FISH analysis (*Figure 2—figure supplement 2A*), where 15 of 100 randomly selected cells scored positive for nuclear HIV-1 proviral DNA for the A77V variant compared to 75 in the case of wild-type HIV-1. Furthermore, the number of viral DNA signals per cell was also lower for the A77V variant (*Figure 2—figure supplement 2B*), consistent with its reduced infectivity. Quantitation of CPSF6 signal intensities associated with individual RTC/PIC stratified for the subcellular localization revealed clear enrichment of CPSF6 on nuclear complexes with an intermediate signal for complexes at the nuclear envelope and no detectable CPSF6 on cytoplasmic complexes (*Figure 4F*, *Figure 4—figure supplement 1C*). Interestingly, this pattern and the observed signal intensities were maintained after CPSF6 knock-down, consistent with our results showing that HIV-1 infection after CPSF6 knock-down preferentially occurs in cells that retain a threshold level of CPSF6. Very low to non-detectable CPSF6 levels were observed on A77V derived complexes (*Figure 4F*, *Figure 4—figure supplement 1C*), consistent with its defect in CPSF6 recruitment. Taken together, our data do not support involvement of CPSF6 in cytoplasmic transport of RTC/PIC to the NPC, but argue for a role of CPSF6 in nuclear import in terminally differentiated macrophages.

## The capsid-binding drug PF74 inhibits HIV-1 infectivity and affects nuclear entry of the RTC/PIC

The small molecule PF-3450074 (PF74) has been shown to block HIV-1 infection by interaction with the viral CA protein (*Blair et al., 2010*; *Shi et al., 2011*). Structural studies revealed that the compound preferentially binds to the assembled CA hexamer, where it targets a pocket that serves as binding site for CPSF6 and for the nucleoporin Nup153 (*Price et al., 2014*; *Bhattacharya et al., 2014*). PF74 has been shown to block HIV-1 reverse transcription at concentrations > 10 μM (*Shi et al., 2011*; *Rasaiyaah et al., 2013*; *Peng et al., 2014*; *Saito et al., 2016a*), most likely by disrupting the integrity of the incoming capsid (*Márquez et al., 2018*). At lower concentrations, the drug has been reported to affect HIV-1 nuclear entry without inhibiting reverse transcription (*Peng et al., 2014*; *Saito et al., 2016a*). To investigate the effects of PF74 in relation to CPSF6-binding of the viral capsid, we performed single-round infections of MDM with wild-type HIV-1 or the CPSF6-recruitment defective A77V variant in the presence or absence of 2.5 μM PF74 and scored infectivity 6d later. PF74 decreased wild-type HIV-1 infection ca. 10-fold compared to the DMSO control (*Figure 3D*) and showed a similar effect on the A77V variant (7-fold reduction; *Figure 3D*). These results suggest that PF74 may exhibit additional effects besides blocking the CA-CPSF6 interaction. Parallel infection experiments were performed with the prototypic CPSF6-binding defective CA variant N74D. In agreement with previous reports (*Schaller et al., 2011*; *Ambrose et al., 2012*; *Rasaiyaah et al., 2013*), we observed a much stronger reduction in infectivity compared to the A77V variant (*Figure 3—figure supplement 2A*), while PF74 had no additional effect on infection by the N74D variant (*Figure 3—figure supplement 2B*).

To determine whether PF74 affects nuclear entry of HIV-1 subviral complexes, we infected MDM with IN.eGFP-labeled wild-type HIV-1 or the A77V variant in the presence of 2.5 μM PF74. PF74 treatment caused a strong reduction in the proportion of nuclear complexes in wild-type HIV-1 infected cells (0.01% corresponding to a single nuclear HIV-1 structure in the presence of PF74 compared to 3.6% nuclear structures in the control; *Figure 4G*, *Figure 4—figure supplement 1D*). Interestingly, we also observed a significant reduction in the proportion of IN.eGFP signals close to the nuclear envelope in the presence of PF74 (8.2% in control vs. 4% in PF74-treated cells; *Figure 4G*, *Figure 4—figure supplement 1D*). In the case of the A77V variant, the proportion of nuclear structures was very low even in the control, and we could thus not conclusively determine whether PF74 had an additional effect on nuclear entry in this case (*Figure 4G*). However, the proportion of IN. eGFP signals close to the nuclear envelope was again higher for this variant compared to wild-type HIV-1 (13,9% vs 8.2%), and this number was also reduced by PF74 treatment (*Figure 4G*, *Figure 4—figure supplement 1E*). These observations indicate that 2.5 μM PF74 exhibits additional effects besides affecting nuclear entry of subviral complexes and may explain why PF74 also inhibits MDM infection by the A77V variant.

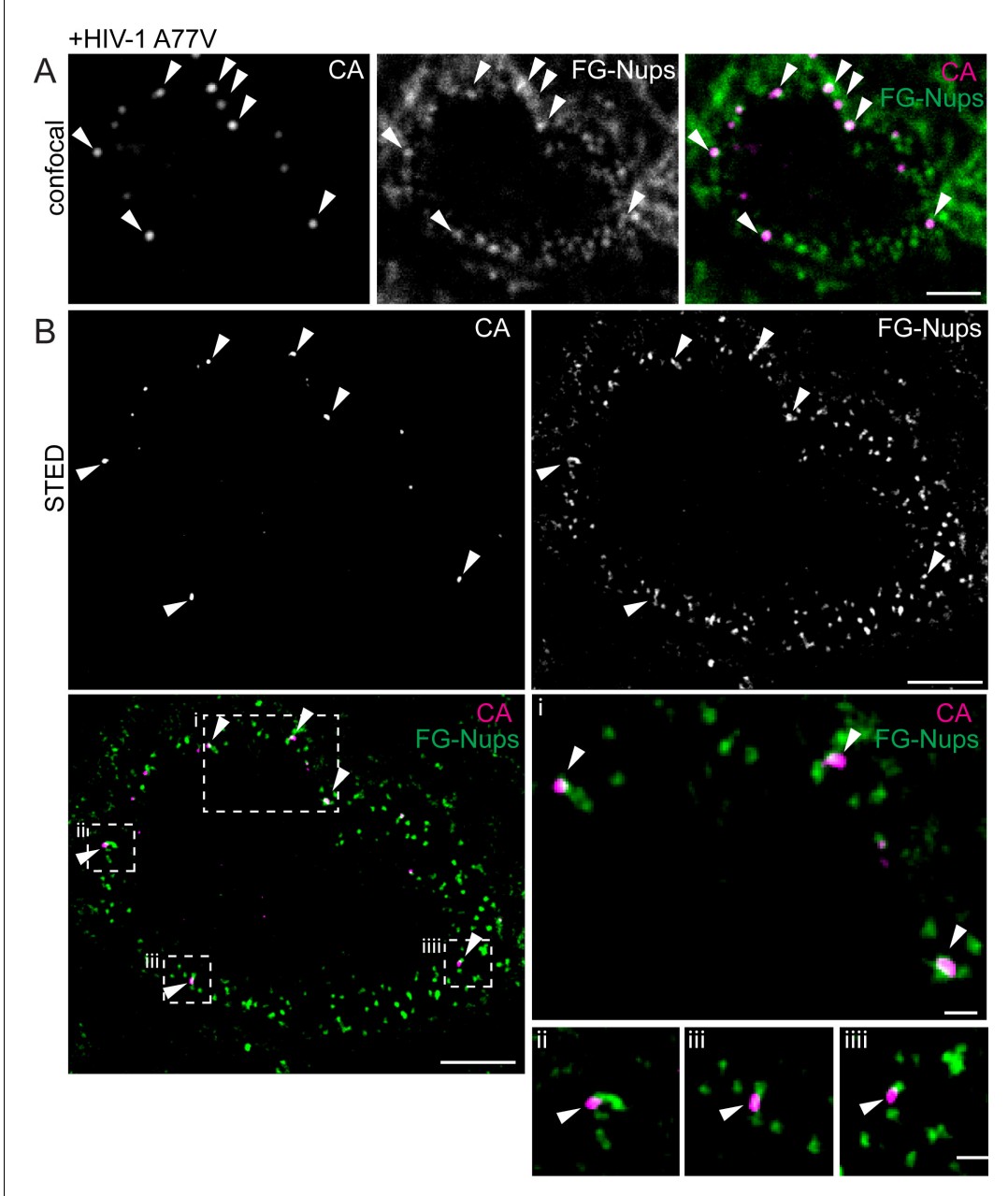

**Figure 5.** Visualization of RTC/PIC of A77V HIV-1 at the NPC using STED nanoscopy. MDM from two donors were infected with 100 ng CA (MOI 14.5) of HIV-1 $_{NL4-3}$ A77V (IN.eGFP) for 60 hr, fixed and immunostained using antibodies against CA (magenta) and NPC proteins (FG repeats; green). (**A**) Confocal images of the nucleus of a representative infected cell. (**B**) STED images of the same infected cell. Enlargements of the four boxed regions are shown to the right of the overlay image. Arrows indicate CA positive objects co-localizing with NPC proteins. Scale bars: 2 μm (confocal images), 500 nm (STED images).

DOI: https://doi.org/10.7554/eLife.41800.022

## HIV-1 RTC/PIC remain arrested at the NPC of infected MDM in the absence of CPSF6

The observation that both, CPSF6 knock-down and the A77V mutation led to accumulation of HIV-1 replication complexes in close proximity to the nuclear envelope suggested that failure to interact with CPSF6 (due to either lack of CPSF6 or lack of CPSF6 binding) may arrest incoming subviral HIV-1 particles at the nuclear pore. To directly test this hypothesis, we performed two-color Stimulated

Emission Depletion (STED) super resolution microscopy of complexes close to the nuclear envelope, achieving a lateral resolution of <50 nm. In order to maximize the number of complexes localized at the nuclear envelope, MDM were infected with HIV-1 (A77V) carrying IN.eGFP at an MOI of 14.5 and fixed at 60 hr. Cells were immunostained for FG-containing nucleoporins to identify NPCs and for HIV-1 CA. Confocal images identified CA-positive complexes close to the nuclear envelope (*Figure 5A*) and STED microscopy revealed that most of these CA-positive structures were directly associated with nuclear pores (34 of 38 CA-positive structures close to the nuclear envelope directly co-localized with NPCs; 89%; *Figure 5B*).

In order to quantify the number of particles arrested at the NPC and obtain comprehensive insight into the relative position of subviral complexes with respect to the NPC, we conducted two-color 3D STED analysis. The nucleus of MDM has a diameter of 6–10 µm along the optical axis. Sampling the entire nuclear volume required the acquisition of super-resolved images in many optical sections, corresponding to 200–300 super-resolved images per nucleus. To minimize bleaching during the acquisition, we implemented a light dose management that specifically activates the STED depletion laser beam to switch off fluorophores in the vicinity of a fluorescent feature to be recorded (*Staudt et al., 2011*). This procedure reduced the number of state transition cycles that a fluorescent molecule undergoes and permitted acquisition of hundreds of super-resolved images instead of a few dozen as typically acquired under standard conditions. This approach enabled us to reconstruct the entire nucleus of a MDM infected with HIV-1 (A77V) for 60 hr, achieving an almost isotropic final resolution of 100 nm x 100 nm x 150 nm (xyz) (*Video 1*). Analysis of these reconstructions revealed that 88% of CA-positive complexes at the nuclear envelope were associated with nuclear pores (408/465, five cells from two independent experiments).

To validate these results, we depleted CPSF6 and subsequently infected MDM with wild-type HIV-1 or the A77V variant carrying IN.eGFP. In addition to CA immunostaining, we performed immunostaining of the nuclear basket protein Nup153, which binds the CA hexamer overlapping with the CPSF6 binding site and has been implicated in HIV-1 nuclear import (*Matreyek et al., 2013*). Samples were analyzed by two-color STED microscopy. Depletion of CPSF6 induced clear accumulation of IN.eGFP and CA double-positive structures at the nuclear envelope of cells infected with wild-type HIV-1 (*Figure 6A*), similar to the phenotype of the A77V variant (*Figures 5,6C*). The vast majority of these structures co-localized with Nup153 (*Figure 6A,C*; STED images), confirming arrest of the HIV-1 replication complex directly at the NPC. Interestingly, line profile analysis of individual wild-type HIV-1 (with CPSF6 knock-down; *Figure 6B*) or A77V subviral particles (*Figure 6D*) revealed partial co-localization of CA signals with the nuclear basket protein Nup153. The main CA signal localized to the cytoplasmic side of the Nup153 signal with a distance between the peak intensities of approximately 70 nm.

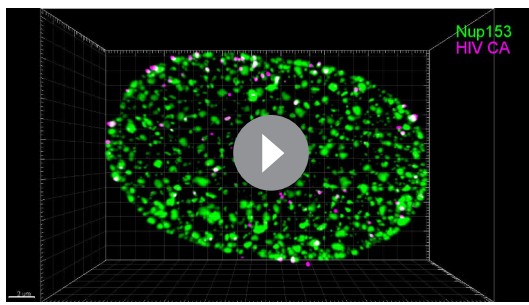

**Video 1.** Related to *Figure 6*. 3D super resolution imaging and co-localization analysis of MDM infected with HIV-1 $_{NL4-3}$ 4059-A77V (IN.eGFP). The video shows a 3D reconstruction of the entire nucleus of an MDM infected with HIV-1 $_{NL4-3}$ 4059-A77V (IN.eGFP) at an MOI of 14.5 for 60 hr. For co-localization analysis both Nup153 and CA signals were modeled as ellipsoids with Z axis = 1.5*X,Y axis. Green – Nup153; magenta – CA; red, CA/Nup153 co-localizing structures.
DOI: https://doi.org/10.7554/eLife.41800.023

## Association of RTC/PIC with CPSF6 occurs at the NPC

Signal intensities for CPSF6 were high for nuclear HIV-1 complexes, much lower for complexes adjacent to the nuclear envelope and undetectable for cytoplasmic structures (*Figure 4F*). This observation would be consistent with initial recruitment of CPSF6 to the subviral structure at the nuclear basket of the NPC (where Nup153 resides) *via* interaction with the hexameric capsid lattice. To characterize the potential recruitment of CPSF6 to HIV-1 replication complexes at the NPC, we performed two-color STED microscopy of MDM following partial depletion of CPSF6 and infection with wild-type HIV-1 or the A77V variant (carrying IN.eGFP). Fixed cells were immunostained for CPSF6 and Nup153. Wild-type HIV-1 infected MDM without CPSF6 depletion exhibited intranuclear HIV-1 replication complexes positive for IN.eGFP and CPSF6 and clearly

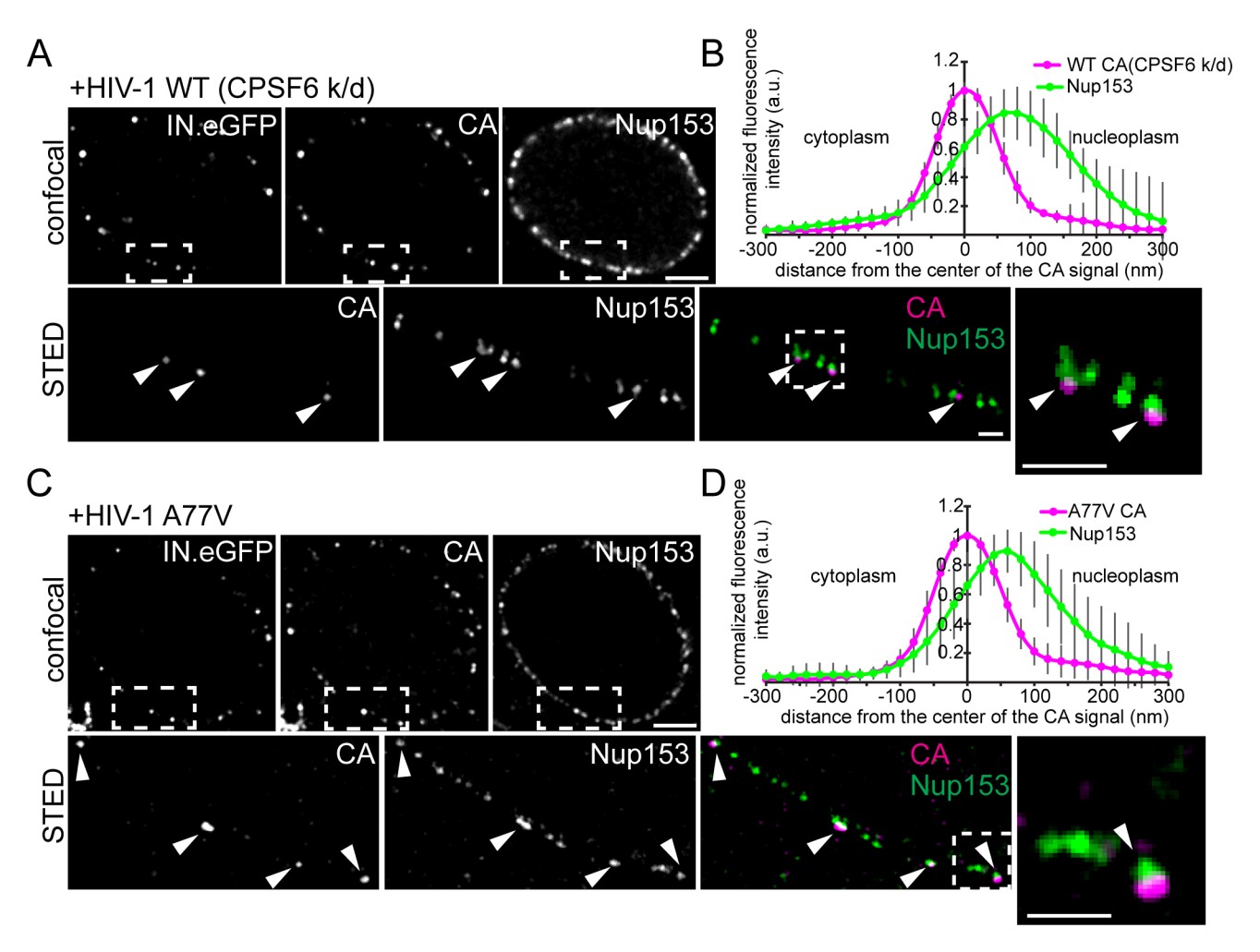

**Figure 6.** HIV-1 subviral complexes arrested at the NPC analyzed by STED nanoscopy. MDM were transduced with AAVs expressing shRNAs targeting CPSF6 (**A, B**) or non-targeted shRNA (**C, D**) as described in Materials and Methods. Subsequently, cells were infected with 100 ng CA (MOI 14.5) of HIV-1 $_{NL4-3}$ WT (IN.eGFP) (**A, B**) or HIV-1 $_{NL4-3}$ A77V (IN.eGFP) (**C, D**). At 60 h p.i., cells were fixed and immunostained against CA (magenta) and Nup153 (green). (**A, C**) Confocal images of the nucleus of a representative infected cell (upper panels) and super resolved images of boxed regions (lower panels). Super resolved images are shown as an overlay on the right (magenta – CA, green – Nup153). Boxed region in the overlayed images (dashed line) are shown as an enlargement. Arrows indicate CA positive objects partially co-localizing with Nup153. Scale bars: 2 µm (confocal images), 500 nm (STED images) (**B, D**) Averaged line profiles from (**A**) or (**C**) of selected CA positive objects (n = 30, from two independent experiments). Error bars represent SD.

DOI: https://doi.org/10.7554/eLife.41800.024

distant from the nuclear envelope (*Figure 7A*). Subviral structures were arrested at the NPC in CPSF6-depleted cells for both wild-type HIV-1 and the A77V variant (*Figure 7B,C*).

Two-color STED microscopy for CPSF6 and Nup153 combined with diffraction-limited detection of IN.eGFP in a third channel revealed clear enrichment of CPSF6 at those Nup153-positive structures (i.e. NPCs) that contained an arrested IN.eGFP-positive HIV-1 replication complex (69 of 80 IN.eGFP-positive structures at NPCs co-localized with a strong CPSF6 signal, 86%; *Figure 7C*). In contrast, a CPSF6 signal was detected on only 16% of NPC-associated subviral structures for the A77V variant (16 of 99 particles; *Figure 7B*), and the CPSF6 signal intensity was much weaker in this case (*Figure 7B*). These observations suggest that CPSF6 is recruited to HIV-1 subviral complexes in a CA-dependent manner when these complexes have reached the nuclear basket. Line profile analysis of individual CPSF6 and Nup153 signals associated with IN.eGFP positive structures further revealed that CPSF6 clusters localize at the nuclear side of the NPC (*Figure 7D*).

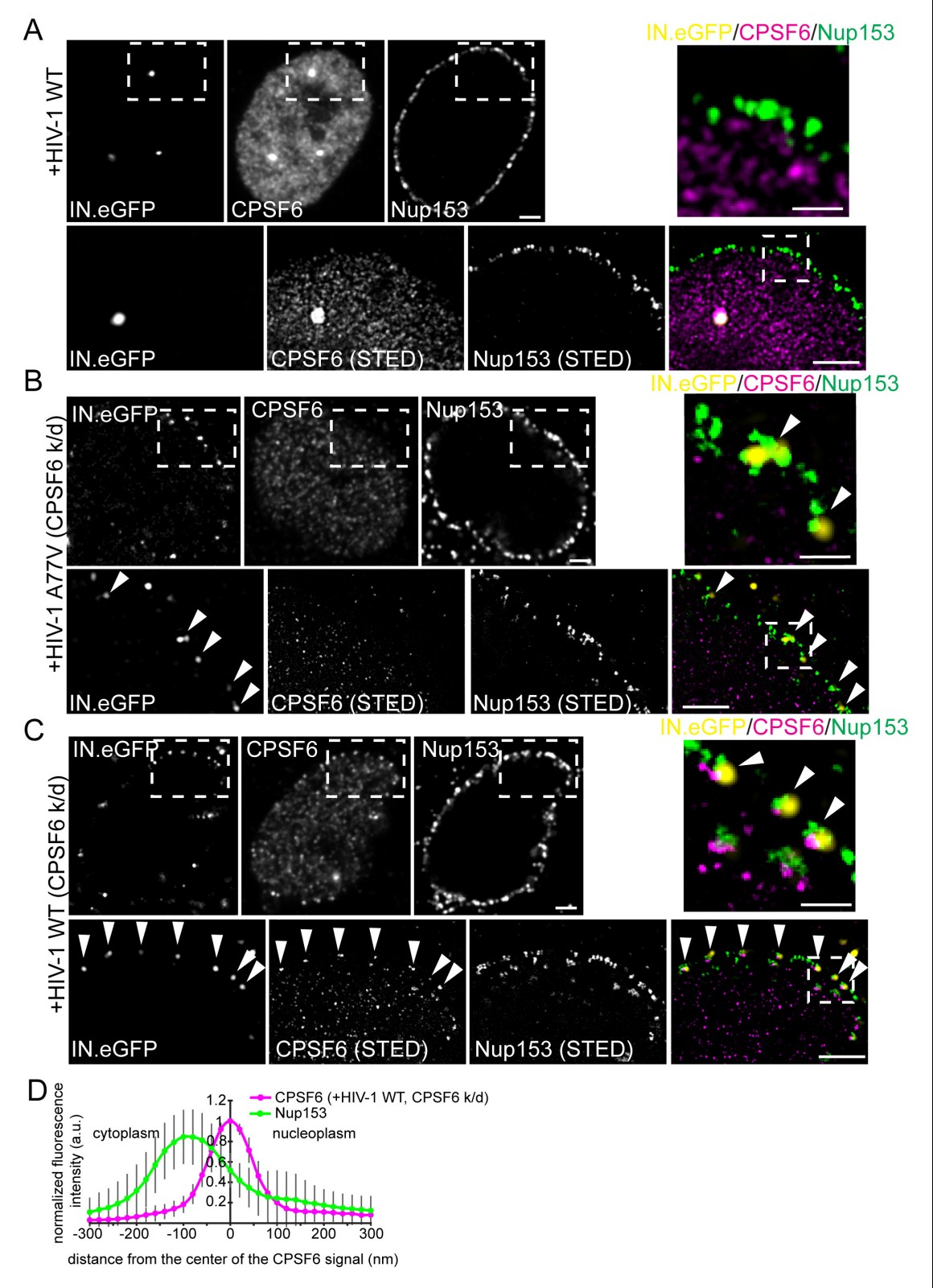

**Figure 7.** Association of CPSF6 and Nup153 analyzed by STED nanoscopy. MDM were transduced with AAV expressing shRNAs targeting CPSF6 (**B, C**) or non-targeted shRNA (**A**) as described in Materials and Methods. Subsequently, cells were infected with 100 ng CA (MOI 14.5) of HIV-1 $_{NL4-3}$ WT (IN. eGFP) (**A, C**) or HIV-1 $_{NL4-3}$ A77V (IN.eGFP) (**B**). At 60 h p.i., cells were fixed and immunostained against CPSF6 (magenta) and Nup153 (green). (**A–C**) Confocal images of the nucleus of a representative infected cell (upper panels) and super resolved images of boxed regions (lower panels). Super

*Figure 7 continued on next page*

Figure 7 continued

resolved images are also shown as an overlay on the right (magenta – CPSF6, green – Nup153, yellow – IN.eGFP (diffraction-limited)). Boxed region in the merged image (dashed line) is shown as an enlargement above the merge. Scale bars: 2 μm (confocal images). 500 nm (STED images). Arrowheads indicate IN.eGFP positive objects at the nuclear membrane. (D) Averaged line profile from (C) of selected IN.eGFP positive objects (n = 30, from two independent experiments). Error bars represent SD.

DOI: https://doi.org/10.7554/eLife.41800.025

## Discussion

HIV-1 infection of all non-dividing cells requires transport of subviral complexes through the intact nuclear pore, and this can also be relevant in dividing cells. Various viral and host cell factors have been implicated in this process (*Matreyek and Engelman, 2013*; *Bin Hamid et al., 2016*; *Yamashita and Engelman, 2017*), but the exact mechanism of nuclear import remains unclear and may be cell-type dependent. We have analyzed nuclear entry of HIV-1 and productive infection in post-mitotic human MDM with a focus on the viral CA and the cellular CPSF6 protein. Applying quantitative microscopy of RTC/PIC, we detected CPSF6 primarily on nuclear complexes; cytoplasmic RTC/PIC were only found to be positive when CPSF6 was artificially targeted to the cytoplasm. No impairment of reverse transcription was observed in the latter case, arguing against the hypothesis that CPSF6 binding to the viral capsid arrests viral DNA synthesis to avoid recognition of viral DNA by DNA sensors (*Rasaiyaah et al., 2013*). In contrast, CPSF6 strongly associated with nuclear HIV-1 complexes, and bright nuclear punctae of CPSF6 could be used to identify subviral complexes following HIV-1 infection or lentiviral vector transduction even against the abundant background of CPSF6 in the nucleus. In agreement with a recent study (*Rasheedi et al., 2016*), we observed that CPSF6 is recruited to HIV-1 replication complexes as component of the $CPSF5^2$-$CPSF6^2$ CF Im complex. This complex is normally involved in pre-mRNA processing, but active transcription was not required for CPSF6 recruitment to HIV-1 PIC. We could also demonstrate that most nuclear subviral complexes co-localized with a signal from LEDGF immunostaining. Our results thus provide direct microscopic support for the model that LEDGF/p75 is recruited to the HIV-1 PIC (reviewed in *Engelman and Singh, 2018*), with the caveat that the antibody used did not allow us to discriminate between the two major isoforms of LEDGF.

Imaging of RTC/PIC revealed that CPSF6 knock-down or the A77V mutation had little or no effect on trafficking of the RTC/PIC to the nuclear envelope, while nuclear import was severely impaired in both cases. HIV-1 RTC/PIC accumulated close to the nuclear envelope under these conditions, indicating that the viral replication complexes were impeded or arrested at this site. RTC/PIC accumulation at the nuclear envelope was also observed for WT HIV-1 infection with normal CPSF6 levels at early, but not at late time points p.i., indicating that transfer across the nuclear envelope is a rate-limiting step as recently suggested (*Burdick et al., 2017*).

In agreement with several recent studies (e.g. *Burdick et al., 2017*), nuclear import of the HIV-1 post-entry complex did not require reverse transcription, but was affected by depletion of CPSF6 or lack of CPSF6 binding (*Chin et al., 2015*; *Ambrose et al., 2012*; *Price et al., 2012*). Few nuclear subviral complexes were observed in these cases and reverse transcription positive complexes were largely retained at nuclear pores. Infection of macrophages was only two- to threefold reduced under these conditions, however. This relatively modest effect may be due to the presence of residual CPSF6, since HIV-1 infection preferentially occurred in cells retaining a threshold level of CPSF6. Furthermore, infectivity was scored on the bulk population of macrophages with incomplete CPSF6 knock-down, while low CPSF6-expressing cells were selected in the imaging experiments.

It should also be considered that HIV-1 RTC/PIC arrested at or close to the nuclear pore in the absence of sufficient CPSF6 recruitment may eventually integrate into chromosomal DNA even without release from the nuclear basket, thus contributing to the observed infection rate. Integration at the NPC and possibly without full release from the nuclear basket is supported by accumulation of HIV-1 proviral DNA in the nuclear periphery for HIV-1 variants with defective CPSF6 binding and for CPSF6 depletion (*Chin et al., 2015*; *Achuthan et al., 2018*). It may also explain the altered integration site profile observed for CPSF6 binding defective HIV-1 variants (*Schaller et al., 2011*; *Saito et al., 2016b*) and upon CPSF6 depletion (*Sowd et al., 2016*; *Rasheedi et al., 2016*; *Achuthan et al., 2018*).

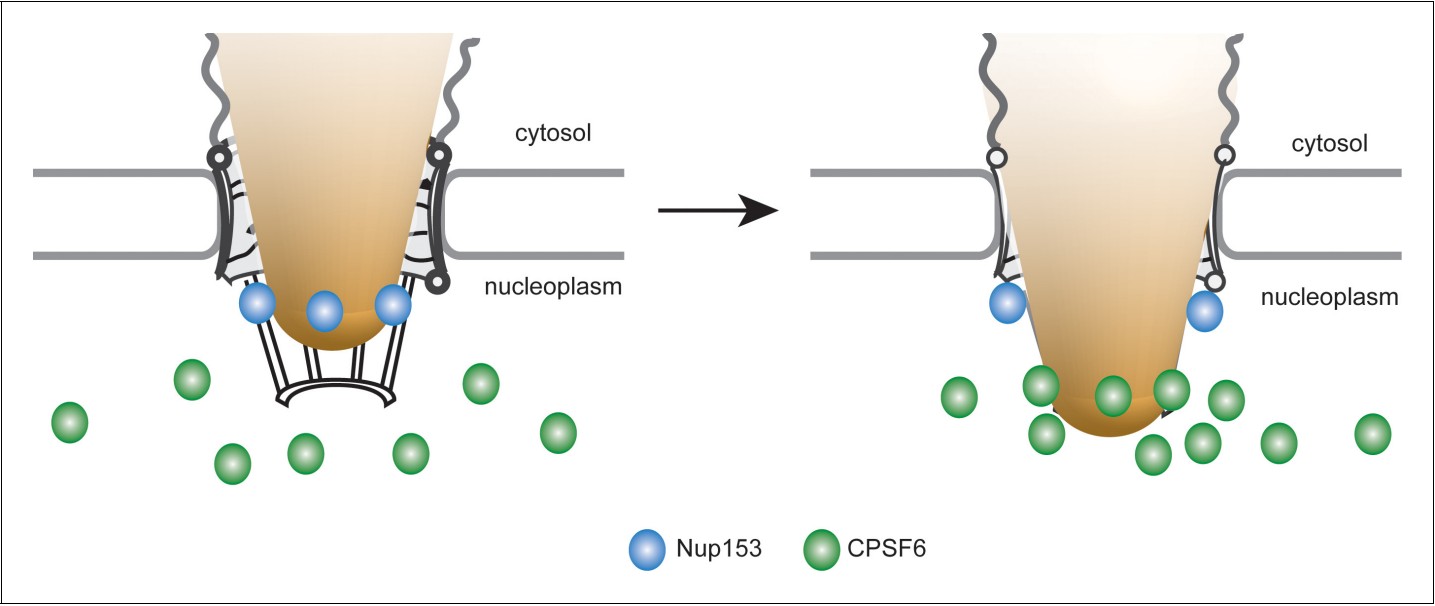

**Figure 8.** Model for the interaction of Nup153 and CPSF6 with the incoming HIV-1 capsid at the NPC of primary macrophages. HIV-1 subviral complexes (depicted as a cone) at the NPC interact with Nup153 (represented as blue dots) at the nuclear basket *via* a capsid-derived structure. CPSF6 (represented as green dots) competitively binds to CA multimers of this structure displacing Nup153 and ultimately releasing the subviral complex from the NPC.

DOI: https://doi.org/10.7554/eLife.41800.026

Both Nup153 and CPSF6 bind preferentially to the CA hexamer (*Price et al., 2014*), indicating that at least some capsid lattice is retained upon transfer through the NPC in MDM. This is consistent with our observation that a strong CA signal is observed on almost all nuclear HIV-1 complexes in these cells. Several recent studies have reported absent or low CA immunostaining on nuclear HIV-1 structures (*Zhou et al., 2011*; *Hulme et al., 2015*; *Mamede et al., 2017*), and we were unable to detect a clear CA signal on nuclear complexes in HeLa-based reporter cell lines as well (*Peng et al., 2014*). In contrast, CA was easily detected on almost all nuclear RTC/PIC of HIV-1 infected MDM, and signal intensities were similar or only slightly reduced when compared to cytoplasmic HIV-1 derived structures detected in the same cell. While this does not prove the presence of an intact HIV-1 capsid, the presence of CA on almost all nuclear complexes, the intensity of the CA signal and the known dependence of CPSF6 binding on the assembled CA hexamer suggest that nuclear HIV-1 replication complexes in infected MDM retain at least a large fraction of their CA coat and expose hexameric binding sites to interacting cellular factors. Whether this complex comprises the entire capsid, a partially uncoated capsid, or a capsid-derived structure stabilized by other factors needs to be investigated by ultrastructural analyses. Stabilization of an incomplete capsid by host factor binding is supported by the recent observation that PF74, binding to the same interface as CPSF6 and Nup153, can stabilize partially disassembled capsids in vitro (*Márquez et al., 2018*). Using a slightly different imaging-based approach, *Francis and Melikyan (2018)* recently reported HIV-1 capsid uncoating at the nuclear envelope, preceding nuclear import of subviral complexes. However, there are pronounced differences in the detection of CA signals on nuclear HIV-1 replication complexes in different cell types (*Peng et al., 2014*), and this apparent discrepancy may simply reflect cell-type specific differences in the mechanism of HIV-1 nuclear import.

Using two-color 2D and 3D STED microscopy, we provide direct proof that HIV-1 RTC/PIC at the nuclear envelope are associated with NPC in almost all cases. Accordingly, lack of CPSF6 interaction arrests or delays the HIV-1 replication complex at or within the NPC. A recent report suggested that CPSF6-capsid interactions allow subviral complexes to penetrate deeper into the nucleus of HEK293T- and primary CD4[+] T-cells for integration, while loss of CPSF6 interaction led to integration into transcriptionally repressed lamina-associated heterochromatin (*Achuthan et al., 2018*). This may be due to integration of arrested PICs adjacent to the nuclear basket or may indicate an

additional role of CPSF6 in intranuclear trafficking of viral PIC. The distance between the peak CA and Nup153 signals in our study was ca. 70 nm, with the CA signal oriented towards the cytoplasmic side. CPSF6, on the other hand, accumulated adjacent to Nup153, but clearly localized towards the nucleoplasm; this CPSF6 clustering was dependent on CA. Taken together, these results suggest that a strongly CA-positive HIV-1 replication complex retaining at least a partial hexameric lattice is arrested inside the pore of the NPC. The central pore is ca. 40–90 nm long, with filaments extending 50–75 nm on both sides (*Beck et al., 2004*). Accordingly, a single HIV-1 capsid or capsid-derived structure of ca. 100 nm in length (*Briggs et al., 2003*) could span the pore. It can concomitantly present numerous CA epitopes at a peak distance of 70 nm from the nuclear basket towards the cytoplasm and induce strong CA hexamer dependent CPSF6 binding on the nucleoplasmic face of the nuclear basket. Tight association with the pore channel and the CPSF6 cluster may partially obscure antibody detection of CA, but the strong CA signal on nuclear complexes indicates that CA is largely retained after NPC passage. CPSF6 recruited to this capsid-derived structure at the nuclear basket might displace Nup153, thereby freeing Nup153 molecules of this NPC to progressively bind to upstream CA hexamers and thus promote nuclear import. Accordingly, abundant nuclear CPSF6 competing Nup153 from the common interface results in the observed large CPSF6 clusters on HIV-1 complexes at the nuclear basket and may eventually cause their release into the nucleoplasm. This model thus proposes consecutive and competitive binding of Nup153 and CPSF6 to the CA lattice on HIV-1 replication complexes as driving force for their nuclear entry; a graphic depiction of the model is shown in *Figure 8*.

We realize that the width of 60 nm at the wide end of the cone shaped HIV-1 capsid (*Briggs et al., 2003*) exceeds the width of the NPC translocation channel of ca. 40 nm (*Beck et al., 2004*), and the capsid-derived structure may be modulated to fit these size requirements. Further studies applying (correlative) cryo-electron microscopy will be needed to define the ultrastructure of this HIV-1 capsid-derived structure at the NPC and to determine whether the NPC structure changes during nuclear translocation of such a large cargo. The experimental system described in this report seems ideally suited to address these important questions, which are relevant not only for HIV-1 biology, but for understanding nuclear transport of large cargo and the flexibility of the NPC in general.

## Materials and methods

### Key resources table

| Reagent type (species) or resource | Designation | Source or reference | Identifiers | Additional information |
|---|---|---|---|---|
| Antibody | Alexa fluor conjugated secondary antibodies | Thermo Fisher Scientific | | IF (1:1000) |
| Antibody | Goat anti-mouse IgG Abberior STAR RED | Sigma-Aldrich; Cat# 52283 | | IF (1:200) |
| Antibody | Goat anti-rabbit IgG Atto 594 | Sigma-Aldrich; Cat# 77671 | | IF (1:200) |
| Antibody | Goat polyclonal HRP anti-rabbit IgG | Jackson ImmunoResearch Labs; Cat# 111-035-144 | RRID:AB_2307391 | ELISA (1:2000) |
| Antibody | Mouse monoclonal anti-hCPSF5 | Sigma-Aldrich; Cat# SAB1404890 | RRID:AB_10739397 | IF(1:100) |
| Antibody | Mouse monoclonal anti-hFG repeats (Nuclear Pore Complex) | Abcam; Cat# ab24609 | RRID:AB_448181 | IF(1:300) |
| Antibody | Mouse monoclonal anti-hLamin A/C | Santa Cruz Biotechnology; Cat# sc-7292 | RRID:AB_627875 | IF (1:100) |
| Antibody | Mouse monoclonal anti-hLEDGF | BD Biosciences; Cat# 611714 | RRID:AB_399192 | IF(1:100) |

*Continued on next page*

Continued

| Reagent type (species) or resource | Designation | Source or reference | Identifiers | Additional information |
|---|---|---|---|---|
| Antibody | Mouse monoclonal anti-hNup153 | Abcam; Cat# ab24700 | RRID:AB_2154467 | IF (1:250) |
| Antibody | Rabbit polyclonal anti-hCPSF6 | Atlas Antibodies; Cat# HPA039973 | RRID:AB_10795242 | IF(1:250) |
| Antibody | Rabbit polyclonal anti-hCPSF7 | Atlas Antibodies Cat# HPA041094 | RRID:AB_10794187 | IF(1:200) |
| Antibody | Rabbit polyclonal anti-HIV-1 CA | In-house | | IF(1:1000) |
| Antibody | Sheep polyclonal anti-HIV-1 CA | In-house | | IF (1:1200) |
| Cell line (*H. sapiens*) | HeLaP4 TNPO3-KD | Z Debyser (University of Leuven, Belgium); *Thys et al. (2011)* | | |
| Cell line (*H. sapiens*) | TZM-bI indicator cells | *Wei et al. (2002)* | RRID: CVCL_B478 | |
| Cell line (*Homo sapiens*) | Human embryonic kidney 293 T cells (HEK293T) | *Pear et al. (1993)* | RRID: CVCL_0063 | |
| Chemical compound, drug | Efavirenz (EFV) | AIDS Research and Reference Reagent Program, Division AIDS, NIAID | | |
| Chemical compound, drug | Flavopiridol | Sigma-Aldrich; Cat#F3055 | | |
| Chemical compound, drug | iQ Supermix | BioRad; Cat#1708860 | | |
| Chemical compound, drug | Maraviroc (MVC) | Sigma-Aldrich; Cat#PZ0002 | | |
| Chemical compound, drug | PF-3450074 (PF74) | Sigma-Aldrich; Cat#SML0835 | | |
| Chemical compound, drug | Raltegravir (RAL) | AIDS Research and Reference Reagent Program, Division AIDS, NIAID | | |
| Commercial assay or kit | Click-iT EdU Alexa Fluor 647 Imaging kit | Thermo Fisher Scientific; Cat#C10340 | | |
| Commercial assay or kit | DIG-Nick translation mix | Roche; Cat#11745816910 | | |
| Commercial assay or kit | InviTrap Spin Universal RNA Mini kit | Stratec; Cat#1060100300 | | |
| Commercial assay or kit | SuperScript III Reverse transcriptase kit | Thermo Fisher Scientific; Cat#18080093 | | |
| Commercial assay or kit | TSA Plus system | Perkin Elmer; Cat#NEL 749A001KT | | |

*Continued on next page*

*Continued*

| Reagent type (species) or resource | Designation | Source or reference | Identifiers | Additional information |
|---|---|---|---|---|
| Recombinant DNA reagent | AAV helper plasmid | D Grimm (University of Heidelberg, Germany) | | AAV helper plasmid expressing rep and cap genes for transducing MDM |
| Recombinant DNA reagent | CPSF6 triple shRNA | This study | | Plasmid expressing three shRNAs targeting CPSF6. For packaging into AAV vector. |
| Recombinant DNA reagent | NS control AAV | K Boerner (University of Heidelberg, Germany); *Börner et al. (2010)* | | Plasmid expressing non-targeted shRNA. For packaging into AAV vector. |
| Recombinant DNA reagent | pEnv-4059 | R Swanstrom (University of North Carolina, USA); *Schnell et al. (2011)* | | Plasmid expressing an R5-tropic HIV-1 Env |
| Recombinant DNA reagent | pHIVSIREN | G Towers (University College London, UK); *Rasaiyaah et al. (2013)* | | Plasmids expressing shRNAs targeting CPSF6 or non-targeting shRNA |
| Recombinant DNA reagent | pMD2.G | D Trono (EPFL, Lausanne, Switzerland) | RRID: Addgene_12259 | Plasmid expressing VSV-G |
| Recombinant DNA reagent | pNL4-3 | *Adachi et al. (1986)* | | HIV-1 proviral plasmid |
| Recombinant DNA reagent | pNL4-3ΔEnv | B Müller (University of Heidelberg, Germany) | | HIV-1 proviral plasmid |
| Recombinant DNA reagent | pNL4-3ΔEnv N74D/A77V | This study | | HIV-1 proviral plasmid |
| Recombinant DNA reagent | pNLC4-3ΔTat | T Müller (University of Heidelberg, Germany) | | HIV-1 proviral plasmid |
| Recombinant DNA reagent | psPAX2 | D Trono (EPFL, Lausanne, Switzerland) | RRID: Addgene_35002 | Lentiviral packaging vector |
| Recombinant DNA reagent | pVAE2AE4-5 | D Grimm (University of Heidelberg, Germany); *Matsushita et al. (1998)* | | Adenoviral helper plasmid |
| Recombinant DNA reagent | pVpr.IN.eGFP | A Cereseto (CIBIO, Mattareo, Italy); *Albanese et al. (2008)* | | Plasmid expressing Vpr.IN.eGFP fusion |

*Continued on next page*

*Continued*

| Reagent type (species) or resource | Designation | Source or reference | Identifiers | Additional information |
|---|---|---|---|---|
| Software, algorithm | Autoquant X3 | Media Cybernetics | RRID:SCR_002465 | |
| Software, algorithm | GraphPad Prism | GraphPad | RRID:SCR_002798 | |
| Software, algorithm | Image J | Image J | RRID:SCR_003070 | |
| Software, algorithm | Imaris 8.1 | BitPlane AG | RRID:SCR_007370 | |
| Software, algorithm | Imspector | Abberior Instruments | RRID:SCR_015249 | |
| Software, algorithm | KNIME | Konstanz Information Miner | RRID:SCR_006164 | |
| Software, algorithm | Volocity | Perkin Elmer | RRID:SCR_002668 | |

## Cell culture

Human embryonic kidney 293 T cells (HEK293T) and TZM-bI indicator cells have been previously described (*Pear et al., 1993*; *Wei et al., 2002*).The HeLaP4-derived cell line stably transduced with shRNA targeting TNPO3 (TNPO3-KD) was kindly provided by Z. Debyser (University of Leuven, Belgium) (*Thys et al., 2011*). Cells were cultured at 37°C and 5% $CO_2$ in Dulbecco´s Modified Eagle Medium (DMEM; Thermo Fisher Scientific, Waltham, USA), supplemented with 10% fetal calf serum (FCS; Biochrom GmbH, Berlin, Germany), 100 U/mL penicillin, and 100 µg/mL streptomycin. For preparation of monocyte-derived macrophages (MDM) human peripheral blood mononuclear cells (PBMC) were isolated from buffy coats of healthy donors by Ficoll density gradient centrifugation. PBMC were seeded in RPMI 1640 medium (Thermo Fisher Scientific) supplemented with 10% heat inactivated FCS and antibiotics for 2 hr at 37°C. Subsequently, non-adherent cells were removed, adherent monocytes were washed, and further cultured in RPMI 1640 containing 10% heat inactivated FCS, antibiotics and 5% human AB serum (Sigma Aldrich, St. Louis, USA) or 50 ng/mL macrophage colony-stimulating factor (M-CSF; Peprotech, Rocky Hill, USA; for experiments performed with AAVs) for 10d until differentiation to macrophages.

## Plasmids

The HIV-1 proviral plasmid pNL4-3 has been described (*Adachi et al., 1986*). Plasmid pNL$_{4-3}$ΔEnv contains a 2 bp fill-in of an NdeI site in the *env* ORF resulting in a frameshift and premature stop codon. The A77V and N74D exchanges in the CA-coding region of *gag* were introduced into pNL4-3 through PCR directed mutagenesis, and transferred into pNL4-3ΔEnv through double digestion with BssHII and AgeI, followed by ligation with the corresponding fragment from pNL$_{4-3}$-A77V or -N74D. Plasmid pNLC4-3ΔTat contains a 31 bp deletion in the first exon of *tat* and was kindly provided by Thorsten Müller (University of Heidelberg, Germany). Plasmid pEnv-4059 expressing an R5-tropic Env protein from a primary HIV-1 isolate (*Schnell et al., 2011*) was kindly provided by R. Swanstrom (University of North Carolina, USA). Plasmid pVpr.IN.eGFP (*Albanese et al., 2008*) encoding a Vpr.IN.eGFP fusion protein with an HIV-1 protease cleavage site between Vpr and IN. eGFP was kindly provided by A. Cereseto (CIBIO, Mattareo, Italy). The adenoviral helper plasmid pVAE2AE4-5 has been described (*Matsushita et al., 1998*). The AAV helper plasmid encoding the rep and cap genes was kindly provided by D. Grimm (Heidelberg University, Germany). The vector for expression of triple short hairpin RNA (shRNA) in AAV was also kindly provided by D. Grimm (Heidelberg University, Germany) and will be reported elsewhere. Into this AAV vector, we inserted three CPSF6 targeting sequences (CPSF6-1: 5´-GCGAAGAGTTCAACCAGGAA-3´; CPSF6-2: 5´-GCCAGAAGACCGAGATTACAT-3´; CPSF6-3: 5´-GGTGGACAACAGATGAAGA-3´) under the control of three different promoters or a single non-silencing (5´-TCGGCGCAGTCTAATTATA-3´) shRNA.

The lentiviral vectors pHIVSIREN expressing shRNAs CPSF6-1 (5´-GCCAGAAGACCGAGATTACA T-3´), CPSF6-2 (5´-GCGAAGAGTTCAACCAGGAA-3´) or non-silencing control (5´-TCGGCGCAGTC

TAATTATA-3′) were kindly provided by G. Towers (University College London, UK) (*Rasaiyaah et al., 2013*). The VSV-G-expressing envelope plasmid pMD2.G and the packaging vector psPAX2 were generated in the lab of D. Trono (EPFL, Lausanne, Switzerland) and obtained through AddGene.

## Antisera and reagents

Rabbit and sheep polyclonal antisera against HIV-1 CA were raised against purified recombinant proteins. Mouse monoclonal laminA/C antibody (sc-7292) was purchased from Santa Cruz (Heidelberg, Germany). Mouse monoclonal antibodies against Nup153 (QE5, Ab24700) and Nuclear Pore Complex proteins (mAb414) were purchased from Abcam (Cambridge, UK). Mouse monoclonal antibody against LEDGF (611714; recognizing an epitope present in the p52 and the p75 isoform of the protein) was purchased from BD Biosciences (Franklin Lakes, USA). Affinity purified antibodies against CPSF5 (mouse, SAB1404890), CPSF6 (rabbit, HPA039974) and CPSF7 (rabbit, HPA041094) were purchased from Sigma Aldrich. Alexa Fluor labeled secondary antibodies were purchased from Thermo Fisher Scientific. Secondary antibodies labeled with Atto or STAR RED dyes for STED were purchased from Sigma-Aldrich. 10 mM stock solutions of Efavirenz (obtained through the AIDS Research and Reference Reagent Program, Division AIDS, NIAID), PF74 (Sigma Aldrich) or Maraviroc (Sigma Aldrich) were prepared in dimethyl sulfoxide and stored at −20°C. A 10 mM Stock solution of Raltegravir (obtained through the AIDS Research and Reference Reagent Program, Division AIDS, NIAID) was prepared in $H_2O$ and stored at −20°C. All chemicals and reagents for transfection and FISH were obtained from standard commercial sources, unless indicated otherwise.

## Virus production and characterization

For production of HIV-1 particles and lentiviral vectors, HEK293T cells were transfected using a standard calcium phosphate transfection. For producing viral particles containing IN.eGFP, pNL4-3 was co-transfected with pVpr.IN.eGFP at a molar ratio of 4.5:1. R5-tropic and R5-tropic tat-defective viral particles were produced by co-transfection of pNL4-3ΔEnv (-WT or -A77V) or pNLC4-3ΔTat, pEnv-4059 and pVpr.IN.eGFP at a molar ratio of 4.5:1:1. Lentiviral vectors for CPSF6 knock-down were generated by co-transfection of pHIVSIREN, pMD2.G and psPAX2 at a molar ratio of 2:1.4:1.4. Supernatants containing HIV-1 or lentiviral particles were collected at 36 h p.t., filtered through 0.45 μm nitrocellulose filters and concentrated by ultracentrifugation through a 20% (w/w) sucrose cushion. Subsequently, particles were resuspended in phosphate buffered saline (PBS) solution containing 10% heat inactivated FCS and 10 mM Hepes pH 7.5, and stored in aliquots at −80°C. Virus titer was determined by titration on TZM-bl indicator cells followed by microscopic quantitation of beta-lactamase expressing cells at 48 h p.i.. Titration on this indicator cell line was used as a reference to determine the virus input for imaging and infectivity experiments in macrophages. Particle-associated RT activity was determined by SG-PERT (SYBR Green based Product Enhanced Reverse Transcription assay) (*Pizzato et al., 2009*). The concentration of CA was measured by an in-house enzyme-linked immunosorbent assay (*Wiegers et al., 1998*). Briefly, ELISA plates were coated with 50 ng of monoclonal anti-p24 antibody from hybridoma cell line 183 clone H12-5C (obtained through the AIDS Research and Reference Reagent Program, Division AIDS, NIAID). Subsequently, wells were blocked with 10%FCS (Biochrom) in PBS, and samples of interest (previously diluted in PBS/0.1% Tween20) were added. Antigen detection was done by addition of an in-house polyclonal rabbit antiserum against CA, followed by the addition of goat antiserum against rabbit immunoglobulin G conjugated to horseradish peroxidase (Dianova, Hamburg, Germany), and detection of enzymatic activity obtained from absorbance readings after adding the substrate tetramethylbezidine (TMB; Thermo Fisher Scientific). As standard, purified recombinant HIV-1 CA of known concentration was used.

For production of AAV vectors, HEK293T cells were transfected with Turbofect (Thermo Fisher Scientific). Cells were co-transfected with AAV helper plasmid encoding *rep* and AAV6- or DJP2-*cap* genes, AAV triple shRNA vector and adenoviral helper plasmid at a molar ratio of 1:1:1. 72 h p.t., cells were collected in PBS and lysed by freeze-thaw cycles in liquid nitrogen and subsequent sonification. Cell debris was removed by centrifugation (16000 x *g*, 10 min, at room temperature) and supernatant containing the AAV particles was stored in aliquots at −80°C.

## Virus infection, Click-labeling, immunostaining and FISH

For imaging of RTC/PIC, TNPO3-KD cells and MDM were seeded in 8-well LabTek (#155411, Thermo Fisher Scientific) or in 8-well LabTek II chamber slides (#155409, Thermo Fisher Scientific) for confocal and STED microscopy, respectively. TNPO3-KD cells were seeded in the presence of 6 µM aphidicolin (Sigma Aldrich) and infected on the following day with HIV-1 (IN.eGFP) at a multiplicity of infection (m.o.i.) of 25 in medium containing 10 µM EdU (Thermo Fisher Scientific) and 6 µM aphidicolin. Cells were pre-incubated at 16°C for 30 min and then shifted to 37°C. After 2 hr, medium was removed and replaced by fresh pre-warmed medium containing 10 µM EdU and incubation was continued at 37°C. To stop infection, cells were washed with PBS and fixed with 4% paraformaldehyde (PFA; Electron Microscopy Sciences, Hatfield, USA) for 30 min at room temperature. Subsequently, cells were washed and permeabilized with 0.5% (vol/vol) Triton X-100 for 15 min. Cells were washed and click-labeling was performed for 30 min at room temperature using the Click-iT EdU-Alexa Fluor 647 Imaging Kit (Thermo Fisher Scientific) following manufacturer´s instructions. For immunostaining, cells were blocked for 30 min with 3% bovine serum albumin (BSA) in PBS and incubated with the primary antibody in 0.5% BSA in PBS for 1 hr at room temperature. Cells were washed and incubated with the corresponding secondary antibody for 1 hr at room temperature in 0.5% BSA. For MDM, cells were infected with 100 ng CA of WT or A77V HIV-1 $_{NL4-3}$ (IN.eGFP) pseudotyped with R5-tropic 4059 Env in the presence of 10 µM EdU. This amount of virus corresponds to an MOI of 14.5, based on the infectious titer determined on TZM-bl indicator cells. Infections with unlabeled virus were performed at an MOI of 3.5 corresponding to ~15 ng CA, unless otherwise indicated. It should be noted that MDM are less efficiently infected than TZM-bl cells, thus yielding lower infection rates, and exhibited strong donor-dependent variability. For infection times longer than 24 hr, Maraviroc (Sigma Aldrich) was added to a final concentration of 5 µM at 24 h p.i. to prevent a second round of infection. For experiments with Efavirenz (EFV) and Raltegravir (Ral), MDM were seeded in the same way and infected with HIV-1 in the presence of 5 µM EFV (Sigma Aldrich) or 5 µM Ral (AIDS Research and Reference Reagent Program, Division AIDS, NIAID). For experiments with flavopiridol, 5 µM flavopiridol (Sigma Aldrich) was added to the medium at 96 h p.i. and cells were incubated for further 12 hr before fixation. Fixation, click-labeling and immunostaining were performed as described above.

Detection of viral DNA with FISH was performed essentially as described (*Solovei and Cremer, 2010*). MDM were seeded in 24-well plates with glass cover slips and infected with 50 ng CA (MOI 8, based on titration on TZM-bl indicator cells) of WT or A77V HIV-1 $_{NL4-3}$ with R5-tropic 4059 Env. 72 h p.i. cells were fixed with 4% PFA in PBS for 10 min at room temperature, washed and permeabilized with PBS/0.5% Triton X-100 for 10 min. After permeabilization, cells were treated with ethylene glycol bis-(succinimidyl succinate), permeabilized again with PBS/0.5% Triton X-100, kept overnight in PBS/20% glycerol, and subjected to five freeze-thaw cycles. Subsequently, cells were treated with 0.1N HCl for 10 min and PBS/0.5% Triton X-100 for 5 min, followed by RNAse A (100 µg/mL in 2x SSC) for 1 hr, washing and storage in 50% formamide/2x SSC overnight. 3 µg of an HIV-1 HXB2 proviral plasmid was labeled by nick translation in the presence of 16-dUTP biotin at 15°C for 3 hr using a nick translation kit (Roche, Basel, Switzerland). The probe was precipitated with ethanol in the presence of Cot-1 and herring sperm DNA and resuspended in 20 µL of hybridization buffer (50% formamide, 10% dextran in 2x SSC buffer). 1 µL probe was used per cover slip, denatured at 95°C for 5 min and kept on ice until incubation with immunostained cells on glass cover slips. Samples were heat-denatured at 80°C for 6 min. Hybridization was performed for 48 hr at 37°C in a water bath. Biotin-labeled hybridized probes were detected using the TSA Plus system (Perkin Elmer, Waltham, USA).

For detection of viral RNA by FISH, MDM were seeded in 8-well LabTeks (#155411, Thermo Fisher Scientific) and infected with 50 ng CA (MOI 8, based on titration on TZM-bl indicator cells) of HIV-1 $_{NL4-3}$4059 or HIV-1 $_{NL4-3\Delta Tat}$4059. At 96 h p.i. cells were treated with DMSO or 5 µM flavopiridol. 12 hr after addition of the inhibitors, cells were fixed with 3.7% formaldehyde in PBS for 10 min at room temperature, washed and permeabilized with 70% ethanol at 4°C overnight. After permeabilization, cells were washed, and immunostained as described above. After immunostaining, cells were washed three times with 10% formamide in Stellaris RNA FISH wash buffer A (Biosearch Technologies, Novato, USA; Cat# SMF-WA1-60) at room temperature for 5 min, followed by hybridization. For hybridization, probe was diluted to a final concentration of 125 nM in Stellaris RNA FISH

Hybridization buffer (Biosearch Tech. Cat# SMF-HB1-10) with 10% formamide, and left incubating overnight at 37°C in humid chamber. After hybridization, cells were washed twice for 2 min with Stellaris RNA FISH wash buffer A with 10% formamide, and subsequently washed three times for 2 min with Stellaris RNA FISH wash Buffer B (Biosearch Tech. Cat#SMF-WB1-20-BS). Stellaris probe for RNA FISH was synthesized by Biocat GmbH (Heidelberg, Germany) using HIV-1 NL4-3 proviral plasmid and labeled with CAL Fluor Red 610 dye.

## CPSF6 knock-down, infectivity assays and time-of-addition experiments

For high throuput analysis of infectivity, MDM (ca. $1 \times 10^4$ cells/well) were seeded in 96-well plates (Costar #3606). CPSF6 knock-down was performed using either AAV vectors (for imaging and infectivity experiments) or lentiviral vectors (for infectivity experiments). MDM were transduced once with lentiviral vectors (4d after induction of differentiation) expressing a non-targeted shRNA (NS control) or two different shRNAs against CPSF6 (K/D). A total of 300mU RT (determined by quantitating RT activity) of lentiviral vectors was used for transduction of each well. 24 hr later, medium was replaced, and cells were left standing for another 7d. For AAV transduction, cells were transduced three times (4, 8, 12d after induction of differentiation) with equal amounts of AAV crude lysates expressing three shRNAs against CPSF6 or a non-targeted shRNA. Seven (when using lentiviral vectors) or 12d (when using AAV) after initial transduction, cells were infected in triplicate with 15 ng CA (MOI 3.5, based on titration on TZM-bl indicator cells) of WT or A77V HIV-1 $_{NL4-3}$ with R5-tropic 4059 Env. To block further entry events and prevent secondary infection, 5 µM Maraviroc (Sigma Aldrich) was added to the medium at 24 h p.i. After 6d, cells were fixed in 4% PFA for 90 min and immunostained with anti-CA (sheep) and anti-CPSF6 (rabbit) antisera, as described above. Additionally, cells were counterstained with Hoechst (Thermo Fisher Scientific). Plates were imaged with a fully automated epifluorescence ScanR screening microscope equipped with the ScanR acquisition software (Olympus Biosystems, Shinjuku, Japan). Images were acquired in the Hoechst, CA- and CPSF6-staining channels using the corresponding excitation and emission filters. The percentage of infected cells was quantified as previously described (*Börner et al., 2010*). Mock-infected wells were used as a negative control to set the threshold. For infectivity experiments with PF74, cells were infected as described above, and a final concentration of 2.5 µM PF74 (Sigma Aldrich) was added together with the virus. For time-of-addition experiments, MDM were seeded in 96-well plates and infected as described above. 5 µM Efavirenz or DMSO was added to the infection together with the virus or every 24 hr for 3 days. 5 µM Maraviroc (Sigma Aldrich) was also added to the medium at 24 h p.i. Fresh medium with inhibitors was supplied after 3d. 6d after infection, cells were fixed in 4% PFA for 90 min, immunostained with anti-CA (rabbit) antiserum and counterstained with Hoechst (Thermo Fisher Scientific). Percentage of infected cells was quantified as described above.

## RNA extraction and qRT-PCR to detect HIV-1 transcripts

MDM were infected as described above in the presence or absence of 5 µM Raltegravir. 24 h p.i. 5 µM Maraviroc (Sigma Aldrich) was added to the medium. At 72 h p.i. medium was replaced by fresh medium with 5 µM Maraviroc (Sigma Aldrich) and 5 µM Ral, if needed. 96 hr after infection, medium was removed, and cells were washed twice with PBS before lysis. RNA was extracted using InviTrap Spin Universal RNA Mini Kit (Stratec Biomedical, Birkenfeld, Germany) following manufacturer´s instructions. cDNA synthesis was performed with the SuperScript III Reverse Transcriptase kit (Thermo Fischer Scientific) following manufacturer´s instructions, using 100 ng of RNA. cDNA was used as a template for detecting HIV-1 transcripts with TaqMan quantitative PCR. PCR conditions were as follows: 1X iQ Supermix (BioRad, Hercules, USA), 900 nM primers and 200 nM probe. Cycling conditions were: 98°C for 3 min, 44 cycles of 98°C for 10 s and 60°C for 40 s, followed by 60 cycles with a ramp rate of 0.5°C/cycle for 5 s each starting at 65°C. Primers for detection of *gag* transcripts used were: Forward, 5´ ACATCAAGCAGCCATGCAAAA 3´, Reverse, 5´ TGGATGCAATCTA TCCCATTCTG 3´, Probe, 5´-FAM- AAGAGACCATCAATGAGGAA-TAMRA 3´. Primers and probe binding to eukaryotic 18S rRNA (VIC/MGB, Thermo Fisher 4319413E) were used in parallel as endogenous control for normalization.

## Microscopy

Multi-channel 3D image series were acquired with a Perkin Elmer Ultra VIEW VoX 3D spinning disk confocal microscope (SDCM) using a 100x oil immersion objective (NA 1.4) (Perkin Elmer), with a z-spacing of 200 nm. Images were recorded in the 405, 488, 561 and 640 nm channels. Images of RNA FISH samples were acquired with a Leica SP8 DLS laser scanning confocal using a 63x oil immersion objective (NA 1.4) (Leica, Wetzlar, Germany), with a z-spacing of 300 nm. Stimulated emission depletion (STED) imaging was performed with a $\lambda = 775$ nm STED system (Abberior Instruments GmbH, Göttingen, Germany), using a 100x Olympus UPlanSApo (NA 1.4) oil immersion objective. Images were acquired using the 590 and 640 nm excitation laser lines. Nominal STED laser power was set to 80% of the maximal power of 1250 mW with 20-30μs pixel dwell time and 20 nm pixel size. For 3D STED data 60% of the STED laser power was used for fluorescence depletion in the Z-axis and RESCue illumination scheme was used to minimize bleaching. Sampling frequency was 30 nm in all three axis (xyz). All STED images shown (except 3D STED images) were linearly deconvolved with a Lorentzian function (fwhm 50 nm) using the software Imspector (Abberior Instruments GmbH).

## Image analysis

To quantify the signal intensity from objects distributed throughout the entire volume of the cells, the data (Z-image series) were reconstructed in the 3D space using Imaris 8.1 (Bitplane AG, Zürich, Switzerland). Acquired images were first deconvolved by Autoquant X3 (Media Cybernetics, Rockville, USA) using Constrained Maximal Likelihood Estimation (CMLE) algorithm with 10 iterations and SNR = 20. Next, the xyz coordinates of objects in IN.eGFP channel were automatically identified using the 'spot detection' module in Imaris. Background was subtracted, and an estimated diameter of 300 nm was used for spot detection. Mean signal intensities from selected spots were measured in the EdU, IN.eGFP, laminA and CPSF6/CA channels.

To determine CPSF6 and CA mean intensities from single cells in infectivity assays, images were processed using the Konstanz Information Miner (KNIME, www.knime.org) and the KNIME image processing plugins (KNIP). A previously described workflow was modified (*Grosse et al., 2017*). Briefly, cellular objects in the Hoechst, CA and CPSF6 channels were identified by background subtraction, automatic global thresholding and connected component analysis. For each individual cellular object with positive Hoechst signal, the mean signal intensities from CPSF6- and CA-staining (in the nucleus and in the whole cell) were calculated.

## Statistical analysis

Data analysis was performed using GraphPad Prism software (GraphPad Software, Inc, La Jolla, USA). Statistical significance was only assessed for sample sizes n > 3. Before assessing statistical significance, a Shapiro Wilk test ($\alpha = 0.05$) was performed to verify normality. Two-tailed non-paired Mann-Whitney test ($\alpha = 0.05$) was used to check statistical significance of non-parametric data. In analyses of the distribution of IN.eGFP objects, a non-paired two-tailed Z-test ($\alpha = 0.05$) was performed to assess statistical significance of two proportions.

## Acknowledgements

We thank Anna Cereseto, Zeger Debyser, Dirk Grimm, Michael Malim, Thorsten Müller, Greg Towers and Ronald Swanstrom for providing cell lines or plasmid constructs. We are grateful to Dirk Grimm for allowing us to use the AAV triple shRNA system, to Manuel Gunkel for technical support on KNIME and to Anke-Mareil Heuser for technical support. We would like to acknowledge the microscopy support from the Infectious Diseases Imaging Platform (IDIP) at the Center for Integrative Infectious Disease Research, Heidelberg. The following reagents were obtained through the AIDS Reagent Program, Division of AIDS, NIAID, NIH: Raltegravir from Merck and Company, Inc, Efavirenz from the Division of AIDS, NIAID and anti-HIV-1 p24 Hybridoma (183-H12-5C) (Cat# 1513) from Dr. Bruce Chesebro.

This work was funded in part by the Deutsche Forschungsgemeinschaft (DFG, German Research Foundation) – Projektnummer 240245660 – SFB 1129 project 5 (HGK) and project 6 (BM), by a

project in SPP1923 (HGK) and by the TTU HIV in the DZIF (VL, KB, KLJ, ML, HGK). DAB was supported in part by a PhD student fellowship from the HBIGS graduate school, Heidelberg.

## Additional information

### Funding

| Funder | Grant reference number | Author |
|---|---|---|
| Heidelberg Biosciences International Graduate School | | David Alejandro Bejarano |
| Deutsches Zentrum für Infektionsforschung | TTU HIV | Vibor Laketa<br>Hans-Georg Kräusslich |
| Deutsche Forschungsgemeinschaft | SFB1129 | Barbara Müller<br>Hans-Georg Kräusslich |
| Deutsche Forschungsgemeinschaft | SPP1923 | Hans-Georg Kräusslich |

The funders had no role in study design, data collection and interpretation, or the decision to submit the work for publication.

### Author contributions

David Alejandro Bejarano, Ke Peng, Conceptualization, Formal analysis, Investigation, Visualization, Writing—original draft, Writing—review and editing; Vibor Laketa, Conceptualization, Formal analysis, Investigation, Writing—original draft, Writing—review and editing; Kathleen Börner, Resources, Investigation, Writing—review and editing; K Laurence Jost, Bojana Lucic, Investigation, Writing—review and editing; Bärbel Glass, Investigation; Marina Lusic, Conceptualization, Resources, Writing—review and editing; Barbara Müller, Conceptualization, Funding acquisition, Writing—original draft, Writing—review and editing; Hans-Georg Kräusslich, Conceptualization, Supervision, Funding acquisition, Writing—original draft, Project administration, Writing—review and editing

### Author ORCIDs

David Alejandro Bejarano (iD) http://orcid.org/0000-0001-7804-0131
Vibor Laketa (iD) http://orcid.org/0000-0002-9472-2738
Marina Lusic (iD) http://orcid.org/0000-0002-0120-3569
Barbara Müller (iD) http://orcid.org/0000-0001-5726-5585
Hans-Georg Kräusslich (iD) http://orcid.org/0000-0002-8756-329X

### Decision letter and Author response

Decision letter https://doi.org/10.7554/eLife.41800.030
Author response https://doi.org/10.7554/eLife.41800.031

## Additional files

### Supplementary files

• Source data 1. Correlation analysis. Correlation between CPSF6 knock-down efficiency and HIV-1 infectivity. Spearman correlation of CPSF6 knock-down efficiency and K/D:NS infectivity ratio from multiple donors.
DOI: https://doi.org/10.7554/eLife.41800.027

• Transparent reporting form
DOI: https://doi.org/10.7554/eLife.41800.028

### Data availability

All data generated or analysed during this study are included in the manuscript and supporting files. Source data files for the plots of Figures 1, 3 and 4 and supplemental material are provided.

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
