## [Decision Letter]

Thank you for submitting your article "HIV-1 nuclear import in macrophages is regulated by CPSF6-capsid interactions at the Nuclear Pore Complex" for consideration by *eLife*. Your article has been reviewed by two peer reviewers, and the evaluation has been overseen by Wes Sundquist as Reviewing Editor and Wenhui Li as the Senior Editor. The following individuals involved in review of your submission have agreed to reveal their identity: Alan Engelman (Reviewer #1); Gregory J Towers (Reviewer #2).

The reviewers have discussed the reviews with one another and the Reviewing Editor has drafted this decision to help you prepare a revised submission.

General assessment:

This manuscript is a tour de force analysis of role of CPSF6 in HIV-1 infection of macrophages, rooted in cutting edge fluorescence imaging. The study follows a previous study from this group which established the methods and suggested that CPSF6 was unlikely to be recruited to HIV cores in the cytoplasm. The novel findings in this manuscript are that: 1) cytoplasmic CPSF6 does not restrict HIV-1 reverse transcription in macrophages, as proposed previously by others, and 2) that the CPSF6-CA interaction, which occurs in close association with the nuclear pore complex (NPC), promotes nuclear entry of capsid-containing cores.

The results shed important new light onto mechanism of HIV-1 nuclear translocation in this experimental system, highlighting the CPSF6-CA binding nexus, and surely will be of general interest to *eLife* readers.

Essential revisions:

1) In general, the study does a nice job of using single cell measurements to manage and interpret poor knock down efficiencies in a very convincing way. However, a longstanding issue with image-based virology is the question of how one knows whether any particular image represents an actual infectious event (which is non-trivial with HIV owing to the known low particle-to-infectivity ratio). The Hope and Melikyan labs have now published procedures that correlate imaged foci with cell infection through reporter gene readouts. In the current paper, authors do not have such a correlation between images and infected cells, and it is therefore very important to clarify which datasets strictly correlate with infectious measures. For example, the authors quantify CPSF6 knockdown and HIV-1 infection Figure 3—figure supplement 1, but do not explain how such measures correlate. In some cases, they would not seem to: e.g., donor 2 had marginal (30%?) CPSF6 depletion, yet the infection reduced 4-fold; donor 4, ~4-fold protein depletion, yet a less than 2-fold infection defect. Perhaps rank analysis (such as Spearman) of all 6 donor responses would reveal a positive correlation? Please conduct the analysis and report the results.

2) A related issue is the MOI in each experiment. For example, when the authors discuss quantifying the localization and CPSF6/CA association of ~8400 HIV-1 complexes in a total of 92 infected MDM at 48h p.i. (third paragraph of subsection “CPSF6 is strongly recruited to nuclear PIC in infected macrophages”), it would be helpful to know the expected MOI (can it be stated here and elsewhere?). This issue is important because readers will want to know how many of the observed complexes are expected to be infectious.

3) Another related issue is the lack of clarity of CPSF6 knockdown levels across the different imaging experiments. Does Figure 3—figure supplement 1 encompass all study-wide donors and CPSF6 knockdowns? If so, please indicate in the figure legends which donor samples were used for that experiment. If imaging experiments used other donors (and knockdowns), then it is important know the extents of CPSF6 knockdown and HIV-1 infection defects in each case.

4) Cells express two major isoforms of LEDGF, p75 and p52, but only p75 interacts with HIV-1 integrase. It would therefore clarify things if "LEDGF" were amended to "LEDGF/p75" throughout. More significantly, the antibody being used detects both p52 and p75 and this should be noted. Unlike the work with CPSF6, the work with LEDGF/p75 lacks specificity controls through use of virus or cellular constructs that are known to be defective for the protein-protein interaction. LEDGIN/ALLINI inhibitors of LEDGF/p75-IN interaction could have been used for this. While the additional work is not critical, it is important to at least soften the tone of conclusions since both p52 and p75 would have been visualized in these experiments.

---

## [Author Response]

Essential revisions:1) In general, the study does a nice job of using single cell measurements to manage and interpret poor knock down efficiencies in a very convincing way. However, a longstanding issue with image-based virology is the question of how one knows whether any particular image represents an actual infectious event (which is non-trivial with HIV owing to the known low particle-to-infectivity ratio). The Hope and Melikyan labs have now published procedures that correlate imaged foci with cell infection through reporter gene readouts. In the current paper, authors do not have such a correlation between images and infected cells, and it is therefore very important to clarify which datasets strictly correlate with infectious measures. For example, the authors quantify CPSF6 knockdown and HIV-1 infection Figure 3—figure supplement 1, but do not explain how such measures correlate. In some cases, they would not seem to: e.g., donor 2 had marginal (30%?) CPSF6 depletion, yet the infection reduced 4-fold; donor 4, ~4-fold protein depletion, yet a less than 2-fold infection defect. Perhaps rank analysis (such as Spearman) of all 6 donor responses would reveal a positive correlation? Please conduct the analysis and report the results.

Overall, the reduction in infectivity by either CPSF6 knock-down or A77V substitution was significant, but relatively modest compared to the strong phenotype observed in the imaging experiments. We attribute this difference mainly to the following reasons, while additional factors may contribute as well:

i) Knock-down efficiency was variable between donors and generally in the range of ca. 60% reduction of staining intensity with a high cell-to-cell variation as shown in the figures. Infectivity was scored on the bulk population of cells, while imaging analyses were done on cells selected for low CPSF6 expression (generally >80% knock-down). Thus, one would expect a stronger phenotype in the imaging compared to the infectivity experiments if the two parameters are correlated. We have clarified this aspect in the revised version of the manuscript.

ii) CPSF6 clustering was observed on HIV-1 RTC/PIC even in CPSF6-low cells and the CPSF6 signal localized towards the nucleoplasm compared to the Nup153 signal. These observations indicate that such subviral HIV-1 complexes have gained access to the nucleoplasm despite being arrested or delayed within or close to the nuclear basket. We speculate that these complexes may be competent for chromosomal integration and thus contribute to the observed infectivity. This hypothesis is consistent with the recent publication by Achuthan et al., 2018, as mentioned in the revised discussion.

To directly address the reviewers´ comments, we have performed Spearman analysis correlating reduction in infectivity by CPSF6 knock-down with the overall knock-down efficiency for the respective donor. To increase sample size, we have now performed an additional infection experiment with three donors applying both AAV-based and LV-based knock-down of CPSF6. The figure in Author response image 1 depicts the ratio of percent infected cells under silencing versus non-silencing conditions (multiplied by 100) against the reduction of CPSF6 signal in the knock-down cell population compared to the non-silencing control.

Despite strong variability between donors and experimental variability for the same donor (with some outliers), a significant correlation was observed (r: -0.6235; p<0.0001). Little or no reduction in infectivity was observed for knock-down efficiencies <30%, while infection is reduced two- to threefold at a knock-down efficiency of 40-60%. This result is consistent with our previous analysis scoring HIV-1 infectivity against CPSF6 signal intensities distributed in quartiles (Figure 3B). Furthermore, single cell analysis suggested a threshold of CPSF6 for efficient HIV-1 infection with few infected cells in the very low CPSF6 population (Figure 3C). We have not added the figure to the revised version since we believe that the two arguments above (which are discussed in the manuscript) are more pertinent for the observed difference.

**Author response image 1. respfig1:** Relative infectivity in CPSF6 knock-down cells normalized to infectivity in non-silencing control cells vs. bulk CPSF6 knock-down efficiency determined for the respective sample. Different symbols represent different donors. Open symbols: LV-mediated knockdown; filled symbols, AAV vector mediated knockdown.

In summary, we conclude that CPSF6 knock-down efficiency correlates with reduction in HIV-1 infectivity in MDM, but this correlation may be mitigated by proviruses integrating at or close to the nuclear basket and not reaching deeper into the nucleoplasm. This phenotype is consistent with our proposed model suggesting CPSF6 accumulating on the incoming subviral complex once it has reached the nuclear basket (Figure 8), but not affecting targeting of the subviral complex to the nuclear envelope or into the nuclear pore.

2) A related issue is the MOI in each experiment. For example, when the authors discuss quantifying the localization and CPSF6/CA association of ~8400 HIV-1 complexes in a total of 92 infected MDM at 48h p.i. (third paragraph of subsection “CPSF6 is strongly recruited to nuclear PIC in infected macrophages”), it would be helpful to know the expected MOI (can it be stated here and elsewhere?). This issue is important because readers will want to know how many of the observed complexes are expected to be infectious.

We thank the reviewers for pointing this out. We agree that it is helpful for the reader to provide MOI instead of ng CA values as indicator of input of infectious virus. We have thus added this information and explained in the Materials and methods and legends how infectivity relates to antigen level (thus providing information on total virus input).

Since infectability of primary macrophages shows a strong donor-to-donor variation, the infectivity of particle preparations was routinely characterized by titration on TZM-bl indicator cells. The MOI values presented for unlabeled and IN.eGFP labeled HIV-1 in the revised version of the manuscript refer to this reporter cell titration.

Experiments determining HIV infection rates on macrophages were performed using unlabeled virus. Since TZM-bl cells are more readily infected than macrophages, an MOI of 3.5 was used in most experiments (unless otherwise indicated), while actual infection rates in macrophages were lower. For imaging experiments, we used a higher MOI of 14.5 in order to maximize detection of intracellular viral complexes. We did not perform parallel infection experiments with the IN.eGFP labelled virus in macrophages, however. It should be noted that the ~fourfold higher MOI corresponds to a ~sevenfold higher amount of virus particle input since the IN.eGFP-complemented virus exhibits a ~twofold reduced specific infectivity (Peng et al., 2014). This information is now given in the revised Materials and methods and the MOI values as well as the antigen amounts are stated in the respective figure legends.

3) Another related issue is the lack of clarity of CPSF6 knockdown levels across the different imaging experiments. Does Figure 3—figure supplement 1 encompass all study-wide donors and CPSF6 knockdowns? If so, please indicate in the figure legends which donor samples were used for that experiment. If imaging experiments used other donors (and knockdowns), then it is important know the extents of CPSF6 knockdown and HIV-1 infection defects in each case.

Figure 3—figure supplement 1 does provide information on all donors with CPSF6 knock-down that were used in the imaging experiments. These were the donors with the numbers 4-7.

While this figure shows the distribution of CPSF6 signal intensities and thus confirms the knock-down, it also reveals the strong heterogeneity in CPSF6 levels even after knock-down. However, imaging experiments allowed selection of CPSF6-low cells for the analysis of HIV-1 RTC/PIC localization and composition. Therefore, providing overall knock-down efficiencies for the respective donor would be misleading since CPSF6-low cells were analyzed in all cases, and knock-down simply increased the population size of CPSF6-low cells.

4) Cells express two major isoforms of LEDGF, p75 and p52, but only p75 interacts with HIV-1 integrase. It would therefore clarify things if "LEDGF" were amended to "LEDGF/p75" throughout. More significantly, the antibody being used detects both p52 and p75 and this should be noted. Unlike the work with CPSF6, the work with LEDGF/p75 lacks specificity controls through use of virus or cellular constructs that are known to be defective for the protein-protein interaction. LEDGIN/ALLINI inhibitors of LEDGF/p75-IN interaction could have been used for this. While the additional work is not critical, it is important to at least soften the tone of conclusions since both p52 and p75 would have been visualized in these experiments.

We thank the reviewers for this comment and have softened the conclusions accordingly. We now state that the antibody binds both isoforms, but that only p75 has been shown to interact with IN. Since the antibody is not specific for the p75 isoform, we have not altered the term to LEDGF/p75 when referring to the imaging results, however. As stated in the review, LEDGF was not the main focus of our work, and we have thus refrained from extensive controls in this case.